# Network mechanisms of ongoing brain activity's influence on conscious visual perception

Yuan-hao Wu[1,6], Ella Podvalny[1,5,6], Max Levinson [1] & Biyu J. He [1,2,3,4] ✉

Sensory inputs enter a constantly active brain, whose state is always changing from one moment to the next. Currently, little is known about how ongoing, spontaneous brain activity participates in online task processing. We employed 7 Tesla fMRI and a threshold-level visual perception task to probe the effects of prestimulus ongoing brain activity on perceptual decision-making and conscious recognition. Prestimulus activity originating from distributed brain regions, including visual cortices and regions of the default-mode and cingulo-opercular networks, exerted a diverse set of effects on the sensitivity and criterion of conscious recognition, and categorization performance. We further elucidate the mechanisms underlying these behavioral effects, revealing how prestimulus activity modulates multiple aspects of stimulus processing in highly specific and network-dependent manners. These findings reveal heretofore unknown network mechanisms underlying ongoing brain activity's influence on conscious perception, and may hold implications for understanding the precise roles of spontaneous activity in other brain functions.

Spontaneous brain activity is energetically expensive, large in amplitude, and richly organized in its network structure across multiple spatiotemporal scales[1–4]. An extensive resting-state fMRI literature has established the individual specificity and clinical relevance of brain networks defined by coherent fluctuations of spontaneous brain activity across brain regions. These networks are stable within an individual across days and task states[5], and their alterations in neuropsychiatric illnesses hold diagnostic and prognostic values[6–8]. However, a major unanswered question currently is how such enormous, constantly ongoing spontaneous brain activity shapes task processing. Answering this question not only would provide a stronger mechanistic foundation for interpreting clinical applications based on resting-state networks but also would help to build a comprehensive framework for understanding brain mechanisms underlying task functions.

After all, brain functions are not carried out in isolation but in the context of a constantly active brain, whose state changes from one moment to the next[9].

A line of early fMRI work showed that prestimulus ongoing brain activity can predict trial-to-trial task performance, including perception[10–12], cognitive control[13] and motor output[14]. However, how prestimulus brain activity combines with bottom-up sensory input to shape behavior remains unknown. This is a nontrivial question because contrary to initial thinking[15], multiple studies have shown that in a vast majority of cases, prestimulus activity nonlinearly interacts with stimulus-evoked brain activity instead of linearly add to it[16–19]. This observation suggests that spontaneous activity's impact on behavior cannot simply be thought of as changing the initial state in, for instance, a linear evidence-accumulation computation. By contrast,

[1]Neuroscience Institute, New York University Grossman School of Medicine, New York, NY 10016, USA. [2]Department of Neurology, New York University Grossman School of Medicine, New York, NY 10016, USA. [3]Department of Neuroscience & Physiology, New York University Grossman School of Medicine, New York, NY 10016, USA. [4]Department of Radiology, New York University Grossman School of Medicine, New York, NY 10016, USA. [5]Present address: The Nash Family Department of Neuroscience, Icahn School of Medicine at Mount Sinai, New York, NY 10029, USA. [6]These authors contributed equally: Yuan-hao Wu, Ella Podvalny. ✉e-mail: biyu.he@nyulangone.org

spontaneous brain activity likely shapes task processing through highly interactive and complex mechanisms[20,21]. This idea is supported by recent magnetoencephalography (MEG) and monkey neurophysiology findings showing that the initial state of the brain, as captured by the multivariate activity pattern before stimulus input, influences the trajectory of cortical activity in a high-dimensional state space following sensory input[22,23]. However, these studies have either left unaddressed the brain regions and networks contributing to the initial state and activity trajectory (due to MEG's poor spatial resolution) or narrowly focused on a single brain region (in the case of monkey neurophysiology). As such, the full extent of ongoing brain activity's influences on the behavior investigated as well as the mechanisms underlying these influences remained unknown.

In the animal literature, the importance of considering prestimulus brain states when explaining sensory-evoked activity and the animal's behavioral performance has also been increasingly recognized[9,24]. Yet, most of the studies in this domain have focused on brain states that correlate with arousal fluctuations (e.g., as indexed by pupil diameter, which provides a window into the state of subcortical arousal systems) or gross behavioral states (e.g., running vs. quiescence) e.g., refs. 25–27. As such, how fast, trial-to-trial changes in prestimulus brain activity impact task processing remains poorly understood. In the primary visual cortex (V1), studies have shown that prestimulus neuronal firing rates influence stimulus encoding in the local post-stimulus activity, as well as detection and discrimination performance[28–31]. However, how these findings relate to downstream neural processing directly underlying behavior and whether prestimulus activity in other brain regions, such as higher-order associative cortices, has similar effects on stimulus-related processing remain unknown.

In this work, we take visual perception as a special case to mount a systematic investigation of prestimulus ongoing activity's involvement in task processing. The powerful influence of spontaneous activity on perception is vividly demonstrated by cases such as dreams and hallucinations, where internal brain dynamics create compelling perpetual images all by themselves[32–34]. In addition, it has sometimes been argued that spontaneous brain activity endows an organism with autonomy and is crucial to the emergence of conscious awareness[35–37]. Thus, understanding the role of ongoing, intrinsic brain dynamics in perception not only would allow us to attain a deeper and fuller understanding of how conscious perception arises[35,38] but also may hold clues about what has gone awry in perceptual abnormalities such as hallucinations[39].

To fill this knowledge gap, we employed whole-brain 7 Tesla fMRI data collected during a threshold-level visual object recognition task in 25 healthy subjects (~4 h data collection/subject spread across 2 sessions) to interrogate how prestimulus brain activity from distributed cortical and subcortical brain regions influence multiple aspects of perceptual behavior, including the sensitivity and criterion of conscious object recognition, and discrimination accuracy in a categorization task. To anticipate our results, we found a diverse set of impacts on perceptual behavior by prestimulus ongoing activity originating from visual cortices and regions of the default-mode network (specifically, ventromedial prefrontal cortex and retrosplenial cortex) and cingulo-opercular network (specifically, the thalamus and anterior insula). We further reveal how ongoing activity in each network shapes different aspects of stimulus-related processing, including the magnitude of evoked activity in cortical and subcortical circuits, its trial-to-trial variability, and stimulus content encoding. These results offer detailed mechanistic understanding of how prestimulus ongoing activity from each network influences perceptual decision-making and conscious recognition.

Together, our results provide unprecedented insights into how spontaneous brain activity from distributed brain networks participates in a set of complex human behavior in the context of conscious visual perception and may hold implications for understanding how spontaneous activity facilitates other intricate brain functions.

## Results

### Paradigm and behavior

Twenty-five human participants performed a threshold-level object recognition task while their brain activity was recorded by BOLD fMRI (7 Tesla Siemens scanner). During the task, they viewed images containing real-world objects presented at individually-titrated liminal contrasts (Fig. 1A). The stimuli set comprised images of four object categories: faces, animals, houses, and manmade objects (Fig. 1B, top). The participant's task was to report the image category and whether they subjectively recognized the image content. The category report was via a four-alternative forced-choice question; in trials where the image was not recognized, participants were instructed to make a genuine guess. For the recognition question, participants were instructed to respond "yes" whenever they saw a meaningful object in the image, even if the object appeared unclear and noisy, but to respond "no" if they saw nothing or only low-level features such as lines or cloud-like abstract patterns. As such, this second question probed the success or failure of conscious object recognition rather than conscious detection of low-level image features (for additional details, see ref. 40), aligning with the established definition in prior studies on object recognition[41–43]. Prior to the main task, an adaptive staircasing procedure was employed to titrate each image's contrast to reach an individual participant's recognition threshold (defined as recognition report being ~50% "yes"), and the same physically identical image would be presented on repeated trials during the main task.

Crucially, our stimuli set included both real and scrambled images (Fig. 1B). Scrambled images were generated by phase-shuffling a randomly selected real image from each category to preserve low-level features that differ between categories while removing any meaningful content, and were presented at the same contrast as the corresponding original image. Participants were not informed about the inclusion of scrambled images. Scrambled image trials thus served as catch trials to evaluate the participant's general tendency to report recognition of a meaningful content. Accordingly, recognition rates for the real and scrambled images constituted hit rate (HR) and false alarm rate (FAR), respectively (Fig. 1C). These two measures were subsequently used to derive recognition-related criterion (c) and sensitivity (d') according to Signal Detection Theory (SDT)[44]. The behavioral pattern and stimulus-triggered neural responses related to the success and failure of recognition were described in detail in a previous paper[40]. Here, we briefly summarize key behavioral findings relevant to the present study.

On average, participants reported $48.0 \pm 2.6\%$ (mean ± SEM) of the real images as recognized. The recognition rate for real images (HR) did not differ from the intended rate of 50% ($W = 126.5$, $p = 0.34$, two-tailed Wilcoxon signed-rank test), and was significantly higher ($W = 315$, $p = 4.1 \times 10^{-5}$) than the recognition rate for scrambled images (FAR, $28.0 \pm 3.1\%$; Fig. 1D).

As expected, participants' categorization behavior was dependent on their recognition outcome (Fig. 1E): Categorization accuracy for recognized real images was generally high ($78.8 \pm 2.2\%$) and significantly higher than the chance level of 25% ($W = 325$, $p = 2.9 \times 10^{-8}$, one-tailed Wilcoxon signed-rank test). When real images were unrecognized, the categorization accuracy dropped to $32.0 \pm 1.9\%$ but remained significantly above the chance level ($W = 251$, $p = 0.002$), consistent with previous studies showing above-chance discrimination accuracy in trials where subjects report a lack of conscious awareness[45–47].

To understand the nature of FAR trials (constituting 28% of scrambled image trials), in which participants answered "yes" to the recognition question despite the image input being devoid of any meaningful content, we analyzed the categorization response patterns

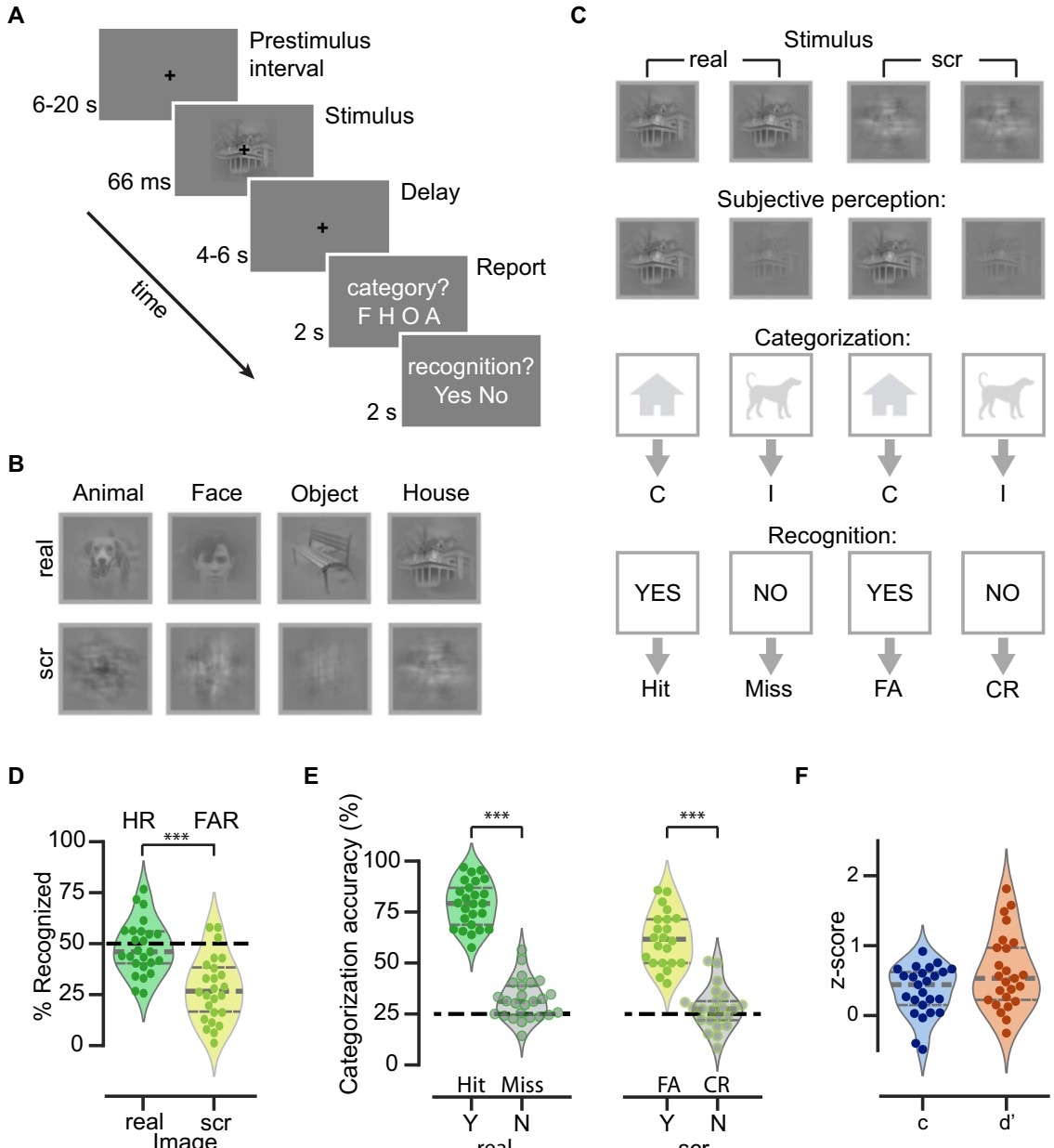

**Fig. 1 | Experimental paradigm and behavioral results. A** Trial structure of the main object recognition task. **B** Example real and scrambled (scr) images from each category. **C** Schematic of trial types and their classification into behavioral metrics. C correct, I incorrect, FA false alarm, CR correct rejection. Categorization responses for scrambled images are coded as correct or incorrect according to the category that the scrambled image was generated from. **D** Percentage of real and scrambled image trials reported as recognized, which are equivalent to hit rate (HR) and false alarm rate (FAR), respectively ($n = 25$, two-sided Wilcoxon signed-rank test, ***$W = 315$, $p = 4.1 \times 10^{-5}$). The black dashed line indicates threshold-level recognition rate (50%). **E** Left: Categorization accuracy for real images grouped by recognition outcome (yes vs no), which are equivalent to hits and misses (two-sided Wilcoxon signed-rank test: ***$W = 325$, $p = 1.2 \times 10^{-5}$). Right: Categorization accuracy

for scrambled images grouped by recognition outcome (yes vs no), which are equivalent to false alarms and correct rejections (two-sided Wilcoxon signed-rank test: $W = 231$, ***$p = 5.96 \times 10^{-5}$). For scrambled images, the categorization accuracy was calculated based on the category of the real image from which the scrambled image was created. The black dashed line indicates chance-level accuracy (25%). **F** Criterion (c) and sensitivity (d') based on the recognition reports (HR and FAR). **A**–**C** Due to copyright limitations, the exact stimuli used in the experiment are not shown in the figure. The face, house, object, and animal images presented here are similar examples obtained from http://www.pexels.com. **D**–**F** Violin plots display the shapes of estimated density probability distributions. The gray, horizontal lines indicate the 25th percentile, median, and 75th percentile. Each circle represents an individual subject.

in these trials. Because the phase scrambling procedure preserved the statistics of low-level features that differed between image categories[48], we scored the accuracy of the categorization responses according to the original images used to generate the scrambled images (e.g., if the participant answered "animal" when the original image was a dog, it was scored as correct). Categorization accuracy for 'recognized' scrambled image (false alarm) trials was $61.7 \pm 2.8\%$. It was significantly higher than that in 'unrecognized' scrambled image

(correct rejection) trials ($27.7 \pm 2.2\%$, $W = 231$, $p = 5.96 \times 10^{-5}$), and significantly above the chance level ($W = 253$, $p = 3.95 \times 10^{-5}$). Thus, low-level features that differed between categories contributed to participants' categorization responses on the false alarm trials, suggesting that the false alarm responses likely reflect genuine false perceptions of meaningful content rather than incorrect button presses[40].

Based on each participant's HR and FAR, we then used the SDT framework[44] to compute the recognition (i.e., detection)-related

criterion (c) and sensitivity (d'). On average, participants had a conservative criterion ($c = 0.35 \pm 0.07$, $W = 29$, $p = 10 \times 10^{-5}$, two-tailed Wilcoxon signed-rank test against 0), and a detection sensitivity of $0.63 \pm 0.16$ ($W = 10$, $p = 2.6 \times 10^{-6}$) (Fig. 1F). A conservative criterion is consistent with previous studies on threshold-level perception[45,49,50].

Finally, to assess whether perceptual behavior changed over the course of the experiment, we divided all trials from each participant into five consecutive time periods and conducted a $1 \times 5$ repeated-measures analysis of variance (ANOVA) across participants, for HR, FAR, c, and d' respectively. This analysis did not reveal any significant effects of the presentation order (HR: $F_{4,96} = 1.68$, $p = 0.16$, $\eta^2 = 0.02$; FAR: $F_{4,96} = 0.42$, $p = 0.79$, $\eta^2 = 0.01$; c: $F_{4,96} = 1.18$, $p = 0.32$, $\eta^2 = 0.01$; d': $F_{4,96} = 0.32$, $p = 0.86$, $\eta^2 = 0.01$), suggesting that all four metrics remained relatively stable throughout the experiment (Fig. S1).

## Prestimulus activity predicts criterion and sensitivity of conscious recognition

We first investigated whether prestimulus ongoing activity influences the criterion and sensitivity of conscious object recognition. To this end, we analyzed fMRI BOLD signals recorded between −2 and 0 s relative to the stimulus onset. Because stimulus onset was always synchronized to the scanner trigger, this amounted to analyzing fMRI activity recorded in the TR immediately before stimulus onset. Because BOLD responses can onset within 800 ms of stimulus input[51], this approach avoids the possibility of post-stimulus activity on the same trial contaminating the analyzed prestimulus activity (as compared to the common approach of shifting the BOLD signal backward in time by ~4 s). We further used a general linear model (GLM) to regress out task-evoked activity from the BOLD signals (for details, see Methods) and the residual was used in the analysis of prestimulus activity. This approach accounted for any lingering activity from the previous trial, which was already mitigated by the relatively long inter-trial interval in our experiment (6–20 s).

For each voxel, we fit two linear mixed-effects models (LMMs) to assess the changes in criterion and sensitivity (defined by recognition reports, see above) as a function of its prestimulus activity (Fig. 2A, see Methods for details). The results yielded from the whole-brain analysis are shown in Fig. 2B, C (displayed at $p < 0.05$, corrected for familywise error rate (FWE) at the cluster level with a cluster-defining voxel-wise threshold of $p < 0.01$).

This analysis revealed significant associations between criterion (c) and prestimulus activity distributed across multiple regions (Fig. 2B; MNI coordinates of all significant clusters available in Table S1). Specifically, we found positive correlations between criterion and prestimulus activity in bilateral visual areas (V3, lateral occipital cortex (LOC), fusiform gyrus (FG), and lingual gyrus (LG)), and in the ventromedial prefrontal cortex (vmPFC). These results suggest that observers are more inclined to report seeing an object, regardless of whether the image contains one when prestimulus activity in these brain regions is low. In contrast, we found negative correlations between criterion and prestimulus activity in the bilateral thalami and the right anterior insula (aInsula). Both regions are the key nodes of the cingulo-opercular (CO) network which also includes the dorsal anterior cingulate cortex (dACC)[52–54]. dACC had a similar trend effect that did not pass the cluster-correction threshold (Fig. S2). These negative correlations indicate that observers are more likely to report seeing an object when the prestimulus activity in the CO network regions is high.

In addition, we found significant positive correlations between sensitivity (d') and prestimulus activity in several brain areas, including two large clusters of voxels located in the vmPFC and retrosplenial cortex (RSC), with the latter cluster extending into the thalamus, as well as three relatively small clusters in the left V3, superior parietal lobule (SPL), and the right LOC (Fig. 2C; see Table S2 for MNI coordinates and Fig. S2 for uncorrected results). These findings indicate that observers are better at discriminating between real (signal) and

scrambled (noise) images when prestimulus activity in these regions is high.

To shed more light on these results, we further investigated prestimulus activity that influenced HR and FAR (Fig. S3, Table S3-4). We found that higher prestimulus activity in visual areas (V3, LOC, VTC) resulted in fewer hits (Fig. S3A) and fewer false alarms (Fig. S3B), consistent with the earlier result showing that higher prestimulus activity in visual areas results in a more conservative criterion (Fig. 2B). Interestingly, high prestimulus activity in the CO network (bilateral thalami, aInsula, ACC) predicted higher hit rates, but had no significant effects on false alarms. In contrast, high prestimulus activity in the vmPFC was primarily associated with lower FAR and largely unrelated to HR (Fig. S3). These results indicate differential prestimulus mechanisms underlying the effects on perception in these brain areas (visual network; CO network; vmPFC), which we will further probe below.

To quantitatively assess brain regions that carried prestimulus activity predictive to both sensitivity and criterion, we overlayed the maps containing significant sensitivity- and criterion-predictive clusters (Fig. S4). Within the vmPFC, the sensitivity- and criterion-predictive clusters had substantial overlap (572 voxels), accounting for 55.7% of the sensitivity-predictive and 47.6% of the criterion-predictive voxels. Moreover, the effects of vmPFC prestimulus activity on both sensitivity and criterion were the largest across the whole brain ($z = 8.16$ for c and $z = 6.58$ for d', see Tables S1 and S2). These results suggest a central and multifaceted role of prestimulus activity in the vmPFC in shaping conscious object recognition. Since vmPFC is a key node of the default-mode network (DMN)[55], this result reinforces a series of recent studies showing the involvement of DMN in visual perceptual tasks[40,56–58] and challenging the conventional wisdom of considering the DMN as a primarily internally-oriented network[59,60].

In comparison, we also observed a small overlapping cluster of 55 voxels in the left V3 (Fig. S4). The relatively small size of the overlap was unsurprising given the finding of a spatially very confined sensitivity-predictive cluster in the visual network (Fig. 2C).

To validate the observed prestimulus activity's effects on behavioral outcomes, we conducted analogous whole-brain LMMs using a different number of trial groups (3 or 7) to identify brain regions where prestimulus activity predicted sensitivity or criterion. As shown in Fig. S5, the results from these analyses are nearly identical to those from the original analysis using 5 trial groups (Fig. 2), suggesting the robustness of our findings. Lastly, to control for the potential effects of head motion, we employed additional LMMs to evaluate whether head motion during the prestimulus period had any predictive influences on conscious recognition (see Methods for details). This analysis did not reveal significant prestimulus head motion effects on any of the behavioral metrics (HR: $\chi = 0.003$, $p = 0.574$; FAR: $\chi = 0.013$, $p = 0.144$; d': $\chi = -0.026$, $p = 0.431$; c: $\chi = -0.023$, $p = 0.149$), rendering head motion an unlikely confounding factor.

Together, these results reveal a multitude of influences of prestimulus ongoing activity on conscious visual recognition, with activity in the visual and cingulo-opercular networks influencing the criterion of recognition (but with opposite effects), and activity in vmPFC influencing both the criterion and sensitivity of conscious recognition. Below, we investigate the detailed mechanisms underlying these effects.

## Prestimulus activity modulates trial-to-trial variability of stimulus-evoked response

Our results thus far suggest that prestimulus activity originating from broadly distributed cortical and subcortical regions exert a diverse set of effects on conscious object recognition, including its sensitivity and criterion—two orthogonal aspects of perceptual behavior. A critical question regards the underlying mechanisms of these modulations: since conscious recognition directly results from stimulus-related

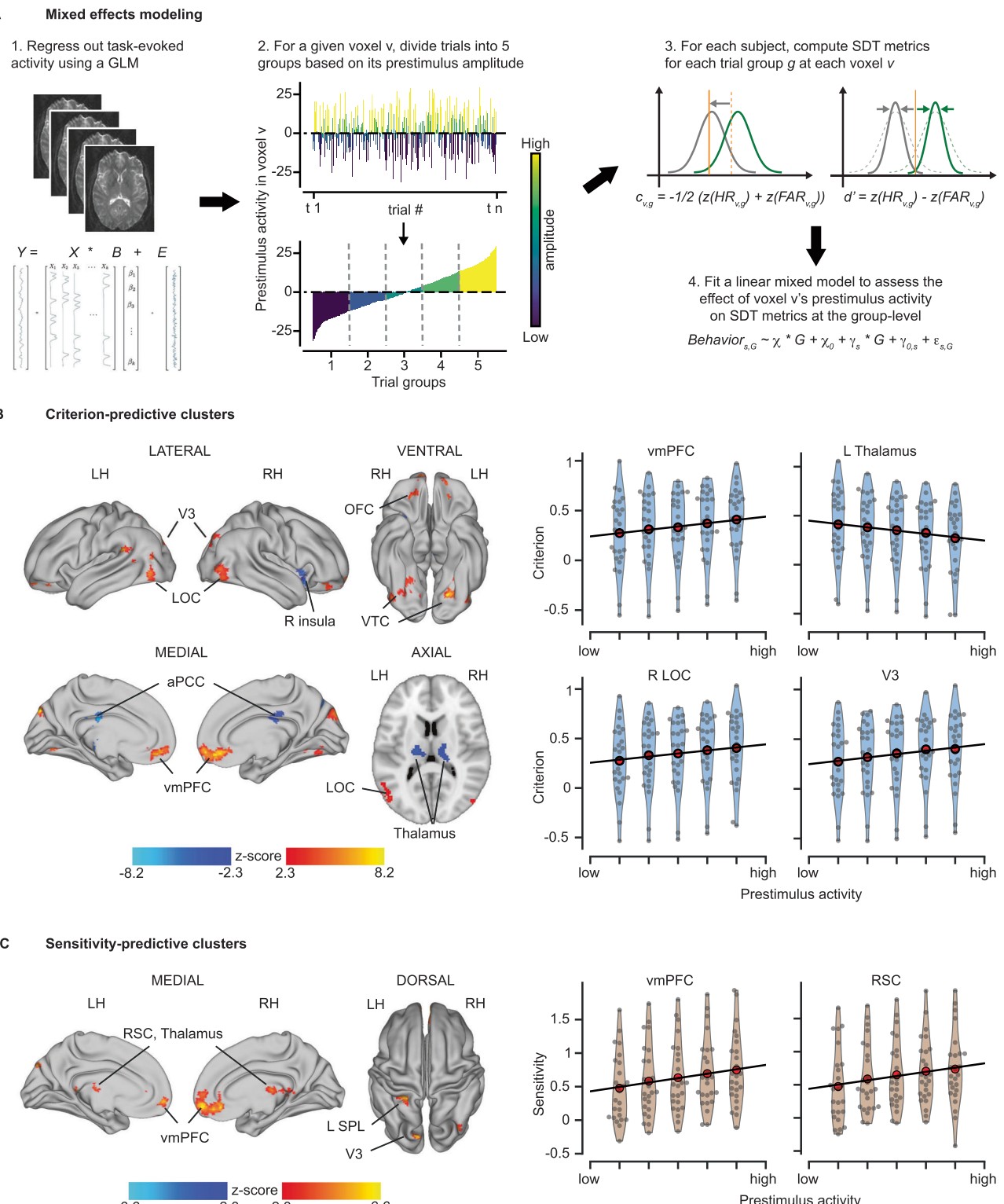

**Fig. 2 | Prestimulus activity influencing the criterion and sensitivity of recognition, respectively. A** Schematic for linear mixed-effect models (LMM) assessing recognition behavior as a function of prestimulus activity. See methods for details. **B** Left: Whole brain statistical maps of prestimulus activity's influence on criterion. Statistically significant positive and negative clusters are shown in warm and cool colors, respectively ($n = 25$, LMM, thresholded at $p < 0.05$, family-wise error (type I, FWE) corrected for cluster size, with a cluster-defining threshold (CDT) of $p < 0.01$). Right: Criterion as a function of the prestimulus activity across voxels within selected brain regions from the left panel. Red circles represent the mean criterion across subjects at each prestimulus activity level. Gray circles represent individual subjects. Blue shapes depict the density estimates of the distributions. The predicted fit of criterion (black line) was estimated based on the mean prestimulus activity across voxels within the ROI using an LMM. **C** Same as B, but for sensitivity (d')-predictive prestimulus activity. aPCC anterior portion of posterior cingulate cortex, LOC lateral occipital cortex, vmPFC ventromedial prefrontal cortex, OFC orbitofrontal cortex, RSC retrosplenial cortex, SPL superior parietal lobule, VTC ventral temporal cortex. LH/RH left/right hemisphere.

**Table 1 | Behavioral effects of high prestimulus activity in each ROI, summarizing results in Figs. 2 and 5**

| Behavioral effects when prestimulus activity is high | Recognition-related criterion (c) | Recognition-related sensitivity (d') | Categorization accuracy |
|---|---|---|---|
| vmPFC | ↑ | ↑ | – |
| Visual network | ↑ | – | ↓ |
| CO network | ↓ | – | ↑ |
| RSC | – | ↑ | – |

processing, how does prestimulus activity from these brain regions modulate stimulus processing differentially to exert a diverse set of effects on perception?

We addressed this question by assessing how different aspects of post-stimulus activity, including its trial-to-trial variability, magnitude, and stimulus encoding, changed as a function of prestimulus activity. To separately shed light on prestimulus activity modulating the criterion and sensitivity of recognition behavior, we defined four regions of interest (ROIs, see Methods for details) based on the analysis shown in Fig. 2B, C. These ROIs included the vmPFC, visual network (bilateral V3, LOC, FG, and LG), CO network (bilateral thalamus, right aInsula), and RSC (for ROI locations, see Fig. S6, left column). The main behavioral effects of prestimulus activity in each ROI are summarized in Table 1, with vmPFC linked to both d' and criterion, visual and CO networks linked to criterion primarily (but in opposite directions), and RSC linked only to d'.

We first examined how prestimulus activity in each ROI affected trial-to-trial variability in the post-stimulus activity. To this end, all trials from each participant were divided into two groups based on each ROI's prestimulus activity according to a median split. For each trial group, we computed the trial-to-trial variability of post-stimulus activity as the across-trial standard deviation of fMRI BOLD signal during the two TRs after stimulus onset (0–4 s), using previously established methods[16]. We then assessed the differences in standard deviation between the two trial groups using a whole-brain voxel-wise contrast, corrected for multiple comparisons using a cluster-correction procedure (for details, see Methods).

Strikingly, vmPFC prestimulus activity modulated trial-to-trial variability of post-stimulus activity across widespread brain areas (Fig. 3A, B). Trial-to-trial variability after stimulus onset in visuomotor areas as well as frontoparietal decision-making circuits was significantly lower when prestimulus vmPFC activity was high. This inverse relationship between vmPFC prestimulus activity and trial-to-trial variability in the post-stimulus responses remained largely stable over the two TRs under investigation. Interestingly, the PCC, a region strongly reciprocally connected with vmPFC[61], was the only brain region showing enhanced trial-to-trial variability when prestimulus vmPFC activity was high, and this positive correlation was only evident in the second TR. Overall, high prestimulus vmPFC activity helped to stabilize post-stimulus activity across widespread brain areas, manifesting as reduced across-trial variability.

In contrast to the widespread vmPFC modulation on trial-to-trial variability, the influence of prestimulus activity originating from the visual network was spatially confined to the early visual cortex (EVC) and high-level visual regions in the ventral temporal cortex (VTC, Fig. 3C, D). When prestimulus activity in the visual network was high, trial-to-trial variability of post-stimulus activity in EVC and VTC was significantly larger. Furthermore, the spatial extent of the effect increased over time, starting from the EVC and posterior VTC in the first TR and progressing to more anterior parts of the VTC and the posterior parietal cortex during the second TR, consistent with the idea that prestimulus activity in the visual network modulates the feedforward signal propagation.

The influences of prestimulus activity in the CO network and RSC were more isolated in space and time. When prestimulus activity in the CO network was high, post-stimulus activity in posterior parietal cortex showed increased trial-to-trial variability during the first, but not the second, TR (Fig. S6C). High prestimulus activity in the RSC predicted reduced trial-to-trial variability in post-stimulus activity in a few small clusters, notably ventral visual regions in the second TR (Fig. S6D).

Because neural activity recorded after stimulus input includes both ongoing and stimulus-triggered activity (which typically interact with each other[16]), we sought to determine whether the effects shown in Fig. 3 are specific to stimulus-related processing or potentially due to the effect of ongoing activity in one region on the variability of ongoing activity in other regions. To this end, we repeated the same analysis by sorting all trials into two groups based on the prestimulus activity in each ROI, but instead of investigating trial-to-trial variability of post-stimulus activity, we tested whether trial-to-trial variability of prestimulus activity differed between these two groups (e.g., whether high vs. low prestimulus vmPFC activity predicted trial-to-trial variability in other regions in the same period). The results of this analysis, plotted in Fig. S6 (−2–0 s period), show that while there are some similarities between the prestimulus (−2–0 s) and post-stimulus (0–2 and 2–4 s) periods, there are also substantial differences. Therefore, although some of the prestimulus activity's effects on post-stimulus trial-to-trial variability might be inherited from the prestimulus period, the majority of the uncovered effects are likely driven by stimulus-triggered processing.

Did prestimulus activity in vmPFC, visual network, CO network, and RSC also influence the magnitude of stimulus-evoked responses? We conducted an additional analysis to address this question (for details, see Methods). The results from this analysis show that CO network prestimulus activity significantly modulated the magnitude of evoked responses in visuomotor areas, including EVC, VTC, and pre/post-central gyrus (Fig. S7A). When CO network prestimulus activity was high, evoked responses in these cortical regions were significantly lower. In contrast, the influence of prestimulus visual network activity was centered on basal ganglia structures, including the caudate and putamen, with high prestimulus visual activity predicting low evoked responses in these subcortical structures (Fig. S7B). Importantly, like visuomotor areas, both caudate and putamen are activated in this task[40], suggesting that its activation is lower when prestimulus visual network activity is high. Interestingly, unlike the prominent modulatory effects of CO and visual networks, prestimulus activity in vmPFC and RSC had minimal impacts on the magnitude of evoked responses.

Taken together, it is of note that in both regions where prestimulus activity modulated recognition sensitivity, vmPFC and RSC (Fig. 2C), prestimulus activity primarily influenced the variability rather than the magnitude of evoked responses: high prestimulus activity resulted in reduced variability in post-stimulus responses. This effect was much more widespread for vmPFC, consistent with vmPFC having a larger impact on d' (see Table S2, z-value in vmPFC: 6.57, in RSC: 3.97). The reduction in across-trial variability may be a mechanism underlying the enhanced recognition sensitivity, a possibility that we will further investigate below. The fact that vmPFC and RSC's effects are spatially distinct may be related to top-down signals from these regions targeting different brain networks (higher-order networks for vmPFC and visual regions for RSC[62]).

By contrast, CO network prestimulus activity primarily influenced the magnitude rather than variability of evoked responses, with higher prestimulus activity in the CO network predicting lower visuomotor activation. Previous work has suggested that the CO network is involved in maintaining tonic alertness[63–65]. In line with this idea, our observation is concordant with previous human and primate findings that during resting state, heightened arousal (as indicated by

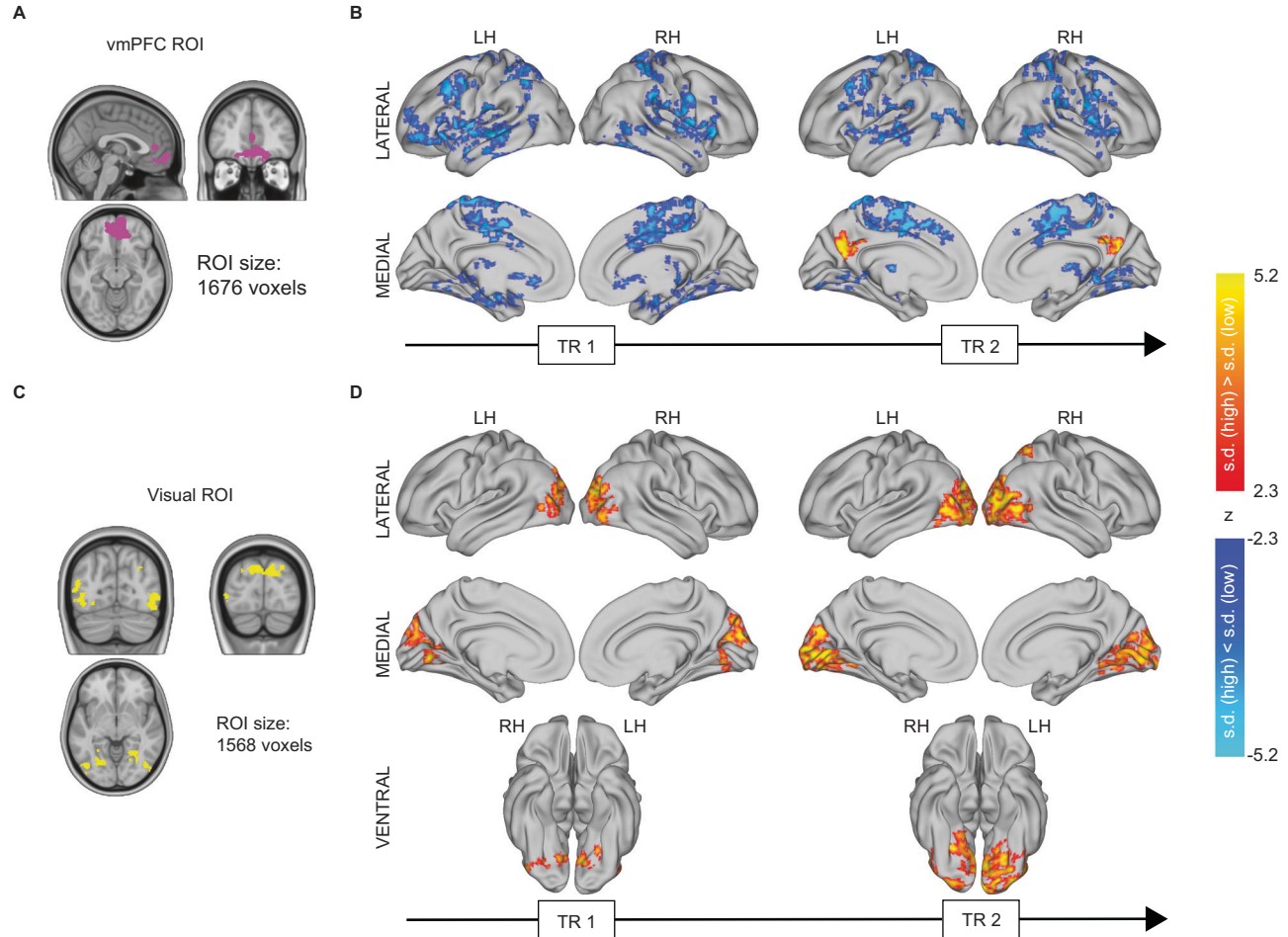

**Fig. 3 | Prestimulus activity's influence on trial-to-trial variability in stimulus response amplitude. A** vmPFC ROI composed of prestimulus activity predictive to criterion and/or sensitivity. **B** Whole-brain statistical maps for differences in post-stimulus trial-to-trial variability between high and low prestimulus vmPFC activity trials from 0 to 4 s relative to the stimulus onset. Voxel clusters with significantly higher trial-to-trial variability (indexed by s.d.) in trials with high prestimulus activity are shown in warm colors, and voxel clusters with significantly lower response variability are shown in cool colors, respectively *(n* = 25, two-sided t-tests, thresholded at *p* < 0.05, FWE corrected for cluster size, CDT at *p* < 0.01). **C** Visual ROI with prestimulus activity predictive to criterion and/or sensitivity. **D** Same as B for the visual ROI. s.d standard deviation, ROI region of interest, TR Repetition time.

eye-tracking) is associated with deactivations across widespread cortical networks centered on visuomotor areas[66,67]. Thus, CO network prestimulus activity might influence recognition behavior through arousal-linked mechanisms, a point we will further discuss below.

Intriguingly, despite the robust influence on trial-to-trial variability in visual regions, prestimulus activity in the visual network was predominantly associated with a criterion effect and had no significant impact on sensitivity (except for a small cluster in V3). Therefore, a more conservative criterion under high prestimulus visual network activity (Fig. 2B) may be due to the downstream network setting a more conservative criterion when incoming signals are noisier (as evidenced by high across-trial variability in the feedforward signals). Importantly, we also observed that high prestimulus visual activity predicted reduced activation in the basal ganglia, and previous animal work found that optogenetic activation in basal ganglia causes the adoption of a more liberal criterion in a visual detection task[68]. Thus, the basal ganglia may be the downstream region in the decision-making network that mediates prestimulus visual network's influence on criterion. It is interesting that the noisier stimulus-related responses (under high prestimulus visual activity) did not impact recognition sensitivity; however, as we will show later, higher prestimulus visual network activity did predispose subjects to have a lower categorization accuracy. Thus, the increased variability in the stimulus-related

responses might primarily lie in the population subspace that separates different categories instead of in the subspace that separates signal (real images) from noise (scrambled images).

## A shared mechanism underlies vmPFC prestimulus activity's impacts on criterion and sensitivity

Our results reveal a central role of vmPFC: Not only did the vmPFC contain the largest clusters of perceptually relevant prestimulus activity, which significantly influenced both sensitivity and criterion of recognition (Fig. 2B, C), but it also stood out as the area with the strongest impact on both of these behavioral metrics across the whole brain (Tables S1 and S2). Could there be a shared mechanism underlying vmPFC prestimulus activity's impacts on sensitivity and criterion? We hypothesized that the reduction in response variability following high vmPFC prestimulus activity (Fig. 3A) might contribute to both an increase in sensitivity and a shift toward a more conservative criterion, which, if true, would provide such a unified mechanism.

To test this possibility, we conducted a simulation within the SDT framework (see Methods). Responses to real and scrambled images were simulated as the internal responses to stimuli containing signal or pure noise. We generated a dataset consisting of 10,000 simulated observers. Internal responses to stimuli containing signal or only noise were sampled from independent, identical Gaussian distributions.

Trial-to-trial variability was modeled as the standard deviations ($\sigma$) of these two distributions. We then simulated how criterion and sensitivity changed with $\sigma$ while keeping the mean responses to targets and nontargets constant. Importantly, since the participants' HR remained constant at around 50% throughout the experiment as intended by the task design and was not affected by the prestimulus vmPFC activity (Figs. S1 and S3A), the decision boundary ($k$) that separated the recognition reports (yes vs no; illustrated by vertical orange lines in Fig. 4B) was set at the mean value of the target distribution for all levels of $\sigma$.

Consistent with our hypothesis, we observed an increased sensitivity and a larger (i.e., more conservative) criterion as the response variability decreased (Fig. 4A, d': $\beta = -0.33$, $p < 0.001$, c: $\beta = -0.17$, $p < 0.001$). As illustrated in Fig. 4B, despite that the decision boundary ($k$) was kept constant at the mean of the target distribution across different $\sigma$ levels (equivalent to keeping HR at 0.5), a reduction in $\sigma$ resulted in a more conservative criterion. This is because a reduction in $\sigma$ led to a decrease in FAR (gray shaded area, $\beta = 0.09$, $p < 0.001$), leading to a more conservative criterion.

Importantly, this simulation shows that a simple reduction in response variability can lead to the full range of behavioral effects observed under increased vmPFC prestimulus activity, including higher d' and c, lower FAR, and a null finding with HR (empirical results in Figs. 2 and S3). These findings thus provide a parsimonious explanation of vmPFC prestimulus activity's influence on visual recognition, and support the possibility that its modulation of response variability may serve as a common mechanism underlying changes in both criterion and sensitivity.

## Prestimulus activity modulates stimulus encoding

We next investigated how prestimulus activity influences stimulus encoding. To this end, we assessed object category information contained in stimulus-evoked activity conditioned on the prestimulus activity level in a specific ROI (Fig. 5A), using the same four ROIs as defined before (vmPFC, visual network, CO network, RSC). Importantly, to avoid bias, logistic regression decoders were trained using stimulus-evoked responses obtained from an independent object category localizer conducted in each subject (for details, see Methods). The trained decoders were then tested on the stimulus-evoked responses from the main task using a whole-brain searchlight approach. Because prestimulus activity influences recognition rates, to avoid the potential confound of unequal behavioral performance, the test set only included recognized real image trials. For each subject, the test set was split into two halves using a median split based on the prestimulus activity of the investigated ROI, and decoding accuracy (obtained from the searchlight analysis) was compared between these two halves in a whole-brain voxel-wise contrast. In addition, to identify meaningful differences between trial groups, the trial group with higher decoding accuracy must also have significantly above-chance decoding accuracy (for details, see Methods).

The results from this data-driven analysis show that prestimulus activity in the visual network significantly impacted stimulus encoding in multiple cortical regions (Fig. 5B). Specifically, low prestimulus activity in the visual network resulted in better category decoding accuracy in the bilateral EVC, VTC, and the right frontal cortex (including middle frontal gyrus and precentral gyrus). There were no brain regions showing the opposite effect. In addition, no significant effects were found when the analysis was conditioned on prestimulus activity from the other three ROIs.

Together, these findings show that prestimulus activity level in the visual network exerts a significant impact on stimulus encoding in both sensory and higher-order associative cortices, reinforcing the earlier interpretation that it shapes bottom-up processing during conscious object recognition. The widespread influences on stimulus encoding by prestimulus visual network activity, along with its influence on recognition-related criterion but not sensitivity, raises the question of

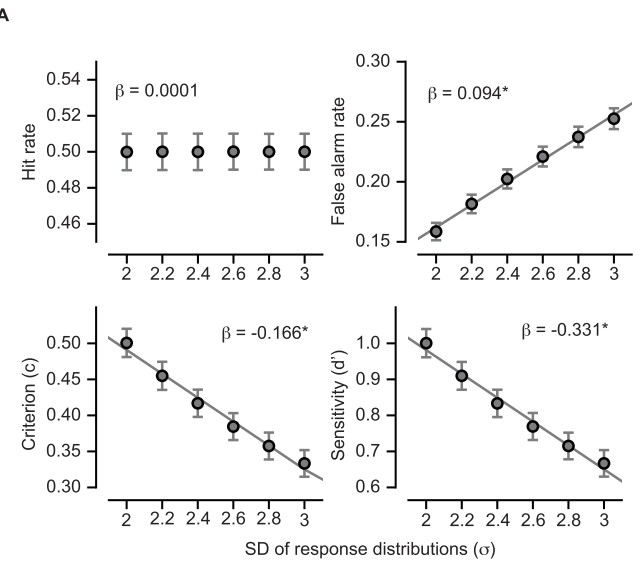

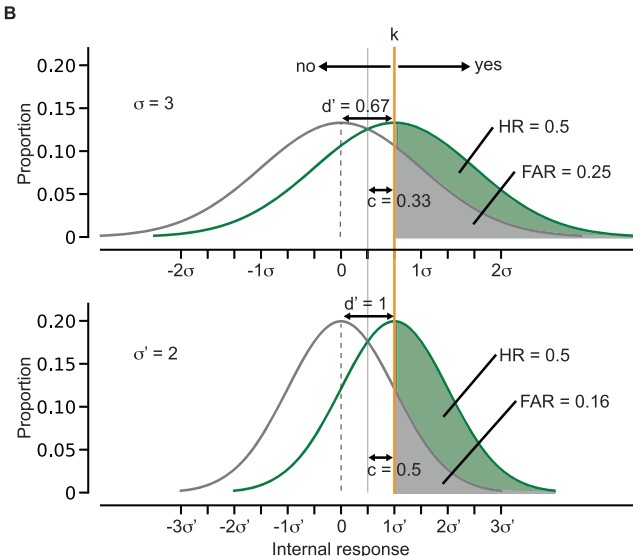

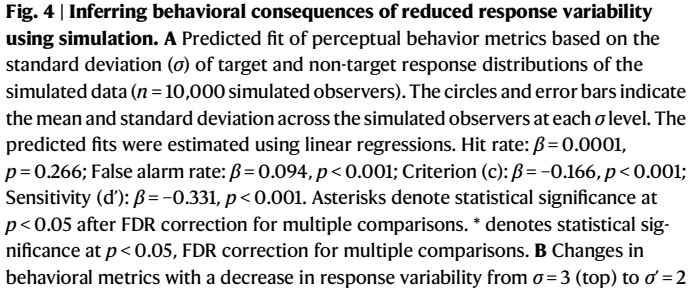

**Fig. 4 | Inferring behavioral consequences of reduced response variability using simulation. A** Predicted fit of perceptual behavior metrics based on the standard deviation ($\sigma$) of target and non-target response distributions of the simulated data ($n = 10,000$ simulated observers). The circles and error bars indicate the mean and standard deviation across the simulated observers at each $\sigma$ level. The predicted fits were estimated using linear regressions. Hit rate: $\beta = 0.0001$, $p = 0.266$; False alarm rate: $\beta = 0.094$, $p < 0.001$; Criterion (c): $\beta = -0.166$, $p < 0.001$; Sensitivity (d'): $\beta = -0.331$, $p < 0.001$. Asterisks denote statistical significance at $p < 0.05$ after FDR correction for multiple comparisons. * denotes statistical significance at $p < 0.05$, FDR correction for multiple comparisons. **B** Changes in behavioral metrics with a decrease in response variability from $\sigma = 3$ (top) to $\sigma' = 2$

(bottom). The curves outlined in green and gray colors represent the distributions of internal responses to target and nontargets, respectively. The orange vertical line indicates the decision boundary ($k$), which separates the yes and no reports. The green shaded area represents HR, which remains unchanged across different $\sigma$ levels. In contrast, FAR, which is displayed by the gray shaded area, decreases with decreasing $\sigma$. Both criterion (c) and sensitivity (d') are measured in the standard deviation (z-score) units (see Methods). c reflects the distance between the decision boundary ($k$) and the point of intersection between the target and nontarget distributions (solid gray line). d' reflects the distance between the means of target and nontarget distributions.

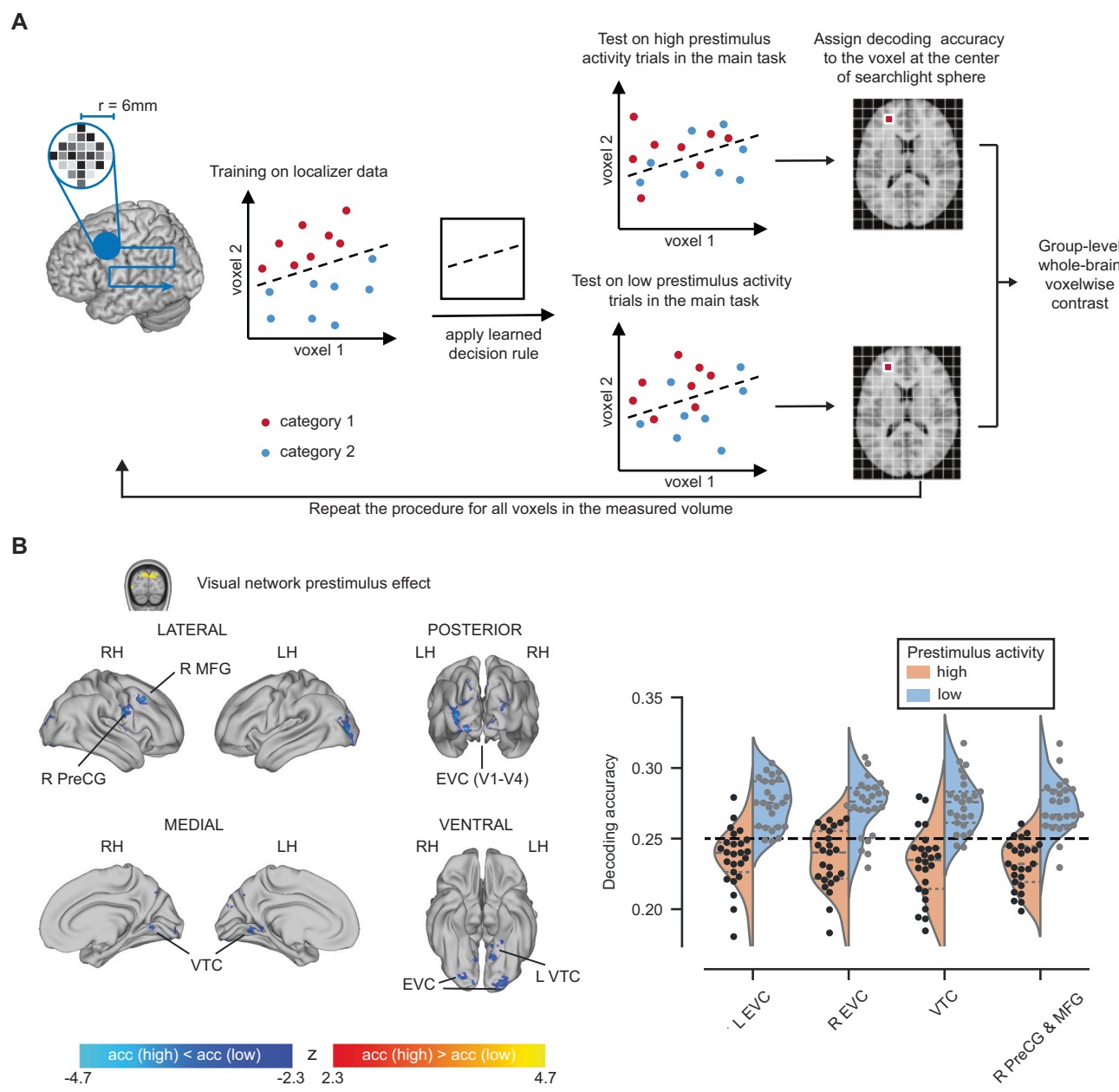

**Fig. 5 | Prestimulus activity's influence on image category encoding.**
**A** Schematic for searchlight decoding. The decoder was fit using data from an object category localizer wherein the object images were presented at high contrast. The decoder was tested using stimulus-triggered responses in the main task (objects presented at a liminal contrast), where the trials were split in two groups based on prestimulus activity amplitude of the investigated ROI. r radius. **B** Left: Whole-brain statistical maps for differences in decoding accuracy between trials preceded by high vs low prestimulus activity in the visual network ROI. Voxel clusters with significantly higher decoding accuracies in high prestimulus activity trials are shown in warm colors, while clusters with higher decoding accuracies in low prestimulus trials are displayed in cool colors. ($n = 25$, two-sided t-tests, thresholded at $p < 0.05$, FWE corrected for cluster size, CDT at $p < 0.01$). Right: Mean decoding accuracy across voxels in each identified cluster when prestimulus visual network activity was high vs. low. Each circle represents one subject. The orange and blue shapes in the background represent the estimated density distributions of the mean decoding accuracies. EVC early visual cortex, MFG middle frontal gyrus, MTG middle temporal gyrus, PreCG precentral gyrus, VTC ventral temporal cortex. L left, R right. LH left hemisphere, RH right hemisphere, acc accuracy.

whether prestimulus visual network activity might modulate participants' categorization behavior, which we investigate in the next section.

**Prestimulus activity modulates categorization behavior**
Lastly, we investigated whether prestimulus activity also influences subjects' categorization behavior. To this end, we adopted a linear mixed-effects modeling approach, similar to that used earlier for assessing prestimulus activity's influences on recognition behavior. This analysis was carried out using the four ROIs defined earlier (vmPFC, visual network, CO network, RSC).

For each participant, all trials were divided into five groups based on each ROI's prestimulus activity, and categorization accuracy was calculated for each trial group. A linear mixed-effects model was then used to assess prestimulus activity's influence on categorization accuracy (see Methods for details). This analysis revealed that categorization accuracy decreased strongly with prestimulus activity in the visual network ($\chi = -0.004$, $p = 0.006$) (Fig. 6). By contrast, categorization accuracy increased strongly with prestimulus activity in the CO network ($\chi = 0.006$, $p = 0.002$). A similar trend was observed in RSC, but it did not surpass the statistical threshold after correction

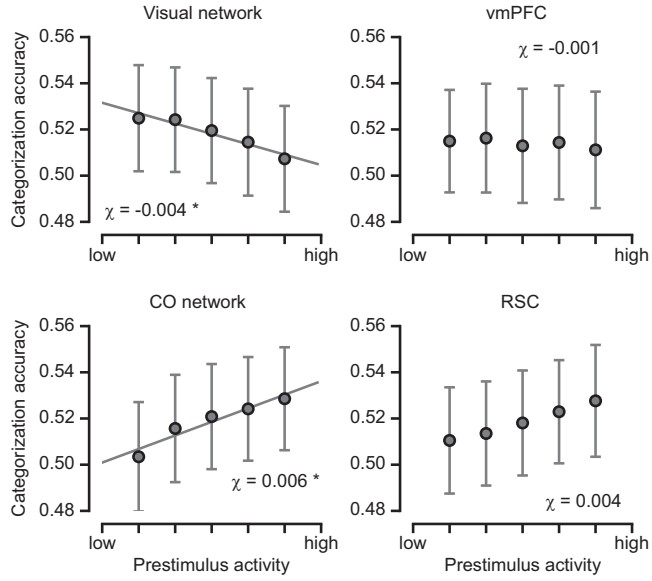

**Fig. 6 | The effect of prestimulus brain activity on categorization behavior.**
Categorization accuracy plotted as a function of the mean prestimulus activity in
the visual network, vmPFC, CO network, and RSC, respectively. The displayed data
points represent the mean categorization accuracy across participants for trial
groups sorted according to the prestimulus activity of each ROI. The error bars
denote SEM. The predicted fit of categorization accuracy was estimated using lin-
ear mixed-effects models. Visual network: $\chi = -0.004$, $p = 0.006$; vmPFC:
$\chi = -0.001$, $p = 0.58$; CO network: $\chi = 0.006$, $p = 0.002$; RSC: $\chi = 0.004$, $p = 0.05$.
*denotes statistical significance at $p < 0.05$ after FDR correction for multiple
comparisons.

for multiple comparisons ($\chi = 0.004$, $p = 0.05$). Prestimulus
activity in the vmPFC did not influence categorization accuracy
($\chi = -0.001$, $p = 0.58$).

Given that both the visual and CO network prestimulus activity
correlated significantly with recognition criterion but in opposite
directions (Fig. 2C), similar to their effects on categorization accuracy,
one might predict a significant correlation between categorization
accuracy and criterion. However, recognition criterion and categor-
ization accuracy had no correlation across subjects ($r = 0.01$, $p = 0.97$),
suggesting that the prestimulus activity's effects on recognition cri-
terion and categorization accuracy were likely independent of
each other.

The inverse relation between visual network prestimulus activity
and categorization accuracy aligns with our earlier finding showing
that low visual network prestimulus activity enhances stimulus pro-
cessing by reducing trial-to-trial variability and improving stimulus
encoding. Interestingly, we also observed a significant positive corre-
lation between the CO network prestimulus activity and subjects'
categorization accuracy, even though we did not observe a significant
effect of CO network prestimulus activity on category decoding (after
correction for multiple comparisons in the whole-brain searchlight
analysis).

## Discussion

In sum, this study reveals the network mechanisms underlying ongo-
ing brain activity's influence on conscious visual perception and per-
ceptual decision-making. Our findings reveal distinct sets of widely
distributed brain regions where prestimulus ongoing activity influ-
ences the sensitivity or shifts the criterion of conscious object recog-
nition, or impacts discrimination accuracy in a categorization task. We
further describe how prestimulus activity modulates multiple aspects
of stimulus-related processing, offering fresh insights into the

mechanisms underlying these behavioral effects. Our key findings are
summarized in Fig. 7 and discussed below.

First, vmPFC, a higher-order brain region that is part of the
default-mode network, was found to play a central role in ongoing
brain activity's influence on conscious recognition. vmPFC was the
only brain region where prestimulus activity significantly influenced
both sensitivity and criterion; moreover, its impact on either beha-
vioral metrics was larger than that of any other brain region. This is a
notable result given that the role of vmPFC is rarely discussed in the
context of conscious visual perception. Our analyses further shed light
on the potential mechanisms of this finding. We found that high
vmPFC prestimulus activity exerted a stabilizing effect on post-
stimulus brain responses across widespread visual and association
cortices. Furthermore, through an SDT simulation, we demonstrated
that a reduction of response variability can result in both enhanced
sensitivity and a more conservative criterion in the context of
threshold-level perception. These results thus provide a unified
explanation for vmPFC's strong and multifaceted influences on
recognition behavior, including both sensitivity and criterion.

Second, we found that high prestimulus activity in visual regions
(including V3, LOC, VTC) resulted in a more conservative criterion for
object recognition (with both lower hit rates and lower false alarm
rates), as well as poorer categorization accuracy. These results can be
explained by our findings in the post-stimulus period: high prestimulus
visual network activity resulted in noisier sensory responses (as evi-
denced by larger trial-to-trial variability), lower activation in the basal
ganglia circuit, and poorer stimulus encoding in both visual and pre-
frontal areas (as evidenced by worse category decoding). Visual net-
work prestimulus activity's influence on categorization accuracy can
be straightforwardly explained by its modulation of stimulus encod-
ing. Regarding the criterion effect, we believe that a parsimonious
explanation is that downstream decision-making circuits adopt a more
conservative criterion when bottom-up stimulus-evoked responses are
noisier. Indeed, the observed inverse relationship between prestimu-
lus visual network activity and the magnitude of evoked responses in
the basal ganglia is consistent with this idea, given that the basal
ganglia circuit has been implicated in setting the perceptual decision
criterion and optogenetic activation of basal ganglia triggers a more
liberal criterion in detection tasks[68,69]. This suggests that the observed
effect of prestimulus visual network activity on basal ganglia activation
might be the mechanism directly underlying its influence on criterion.
An interesting, unresolved question is why the noisier sensory
responses did not lead to reduced recognition-related $d'$. One possi-
bility is that the increased response variability primarily lies in the
subspace of population coding that is relevant for category repre-
sentation instead of the dimension that separate signal from noise (as
illustrated in Fig. 7, bottom-left panel); this intriguing possibility
remains to be tested by future studies.

Third, we observed that high prestimulus activity in the cingulo-
opercular (CO) network—specifically, the bilateral thalami and the
right anterior insular—resulted in a more liberal criterion in the
recognition task as well as better categorization accuracy. The main
effect of high prestimulus CO network activity on post-stimulus pro-
cessing was reduced evoked responses in visuomotor regions. These
results are consistent with a postulated role of the CO network in
regulating tonic alertness[13,65], a point that we will discuss in
detail below.

In what follows, we discuss the implications of these findings, their
relations to prior work, and questions raised by them.

Our findings suggest that the vmPFC prestimulus activity facil-
itates conscious object recognition by stabilizing post-stimulus evoked
responses—an observation reminiscent of earlier findings on reduced
variability during conscious perception[22,70]. Considering the vmPFC's
position at the apex of the cortical hierarchy[71], it is plausible to assume

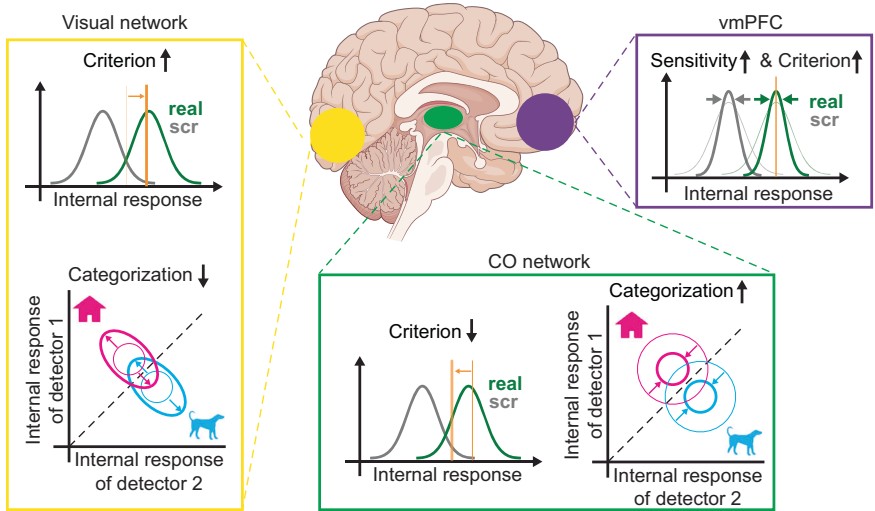

**Fig. 7 | Summary of the effect of prestimulus activity on perceptual processing and perceptual behavior.** Yellow, green, and purple colors rendered on the brain indicate the visual network, CO network, and vmPFC, respectively. Individual graphs summarize the changes associated with enhanced prestimulus activity in each brain region. Under high prestimulus activity in the vmPFC, we observed higher d' and a more conservative criterion in the recognition task and reduced variability in distributed sensory-evoked activity. Under high prestimulus activity in the visual network, we observed a more conservative criterion in the recognition task and lower categorization accuracy, as well as higher neural variability and worse stimulus encoding. Under high prestimulus activity in the CO network, we observed a more liberal criterion in the recognition task and enhanced categorization accuracy.

that this effect involves top-down processing. Among the various higher-level functions attributed to vmPFC[72], we propose that our vmPFC findings may be related to its role in integrating incoming sensory inputs with prior knowledge of the sensory environment[73]. Indeed, previous studies showed that vmPFC activity reflects prior-guided stimulus disambiguation in a visual perception task[56,74]. Accordingly, one possibility is that the state of high vmPFC excitability may facilitate the integration of prior knowledge learnt from lifelong experiences with the sensory input in the service of recognition behavior. The resulting reduction of trial-to-trial variability may serve as a mechanism that aids in disambiguating environmental signals and improving information processing efficiency across the cortex[17,75,76]. This idea is also compatible with previous work suggesting vmPFC's role in generating predictive signals to facilitate the accumulation of sensory evidence to support object recognition[58], as well as vmPFC's established role in maintaining schematic information[77] and cognitive map[78] in the service of flexible behavior.

Therefore, our vmPFC findings provide potential bridges between several traditionally separate research fields, including visual recognition, schema, and cognitive maps, as well as predictive processing which is important in both perceptual and cognitive decision-making. It is of note that vmPFC's central role in conscious recognition was revealed by a data-driven analysis on prestimulus ongoing activity's influence on perceptual behavior (e.g., our previous work focused on post-stimulus activity in this task did not reveal an important role for vmPFC[40]). This is consistent with the idea that ongoing activity facilitates top-down predictive processing[79].

Interestingly, a previous monkey neurophysiological study using a threshold-level detection task concluded that prestimulus neuronal firing rates in the dlPFC predicts detection criterion[31]. By contrast, using whole-brain fMRI in humans, we did not find any sensitivity- or criterion-predictive prestimulus activity in the dlPFC (Fig. 2). Other than differences in species and recording modalities, differences in the task paradigms likely contributed to this discrepancy. While the monkeys in that study detected a low-level visual target (circle) presented in the periphery, in our study human subjects detected the presence of a meaningful visual content (e.g., a house). Therefore, these two studies targeted visual detection vs. object recognition,

respectively. In addition, the authors of ref. 31 suggested that the dlPFC neurons they recorded from were involved in eye movement planning, so they may be especially relevant for visual detection in the periphery. By contrast, our stimuli were presented at fovea.

Our findings that low prestimulus activity in the visual network predicted higher categorization accuracy in the behavioral output and improved stimulus category encoding in the post-stimulus activity fit well with recent monkey neurophysiological studies[28,29]. Recording in V1, these studies showed that lower prestimulus neuronal firing rates predict lower correlated variability and sharper tuning after stimulus input, along with better discrimination performance. In addition, a recent human intracranial study showed that in high-level visual cortex, lower prestimulus activity, as measured by broadband field potential power, precedes faster reaction times and better category encoding in the neural activity[80]. However, these earlier studies did not probe conscious recognition or downstream brain mechanisms, and our findings herein fill in these gaps, by showing that high prestimulus visual network activity results in noisier feedforward sensory-evoked responses as well as lower activation in the basal ganglia circuit, leading to a more conservative criterion during conscious recognition. In addition, we found that prestimulus visual network activity's influence on stimulus encoding extends beyond the visual network to downstream decision-making circuits in the prefrontal cortex, filling in a missing link between previous neural observations in the visual cortex and behavior.

A line of human EEG work showed that prestimulus alpha power primarily influences the criterion but not sensitivity in detection tasks, with lower prestimulus alpha power predicting a more liberal criterion[50,81–83]. On the surface, these results seem contradictory to our finding of lower prestimulus fMRI activity in the visual network being associated with a more liberal criterion. This is because high alpha power in the visual cortex is typically considered to be a state of high inhibition[84]. However, a previous monkey study employing simultaneous fMRI and local field potential (LFP) recordings showed that spontaneous fMRI signal is positively correlated with locally measured alpha power in the visual cortex[85], suggesting that our result is in fact consistent with these previous EEG studies. The relationship between fMRI signal and neurophysiological activity—including LFP in different

frequencies and neuronal firing—is highly complex[86–88], and the detailed neurophysiological and circuit-level mechanisms underlying these findings await future investigation using methodologies allowing access to specific circuit elements and neuronal types (e.g., ref. 89).

The CO network regions, including the thalamus and aInsula, stood out as the only regions exhibiting an inverse relationship between prestimulus activity and criterion, with high prestimulus activity leading to a more liberal criterion. (dACC, another key node of the CO network, had a similar trend effect with criterion, and a significant positive correlation with hit rate; see Figs. S2 and S3). In addition, high prestimulus CO network activity led to enhanced categorization accuracy. These findings shed light on earlier results showing that high prestimulus CO network activity facilitates near-threshold visual, auditory, and somatosensory detection[10,11,90], and suggest that these behavioral effects were likely due to a change in criterion.

CO network constitutes one of the core cognitive control systems[52] and has been implicated in functions such as task set maintenance and sustained vigilance[54]; it overlaps with the salience network[91,92]. Importantly, growing evidence supports the role of the CO network in the maintenance of tonic alertness[63–65]. A recent study[93] using intracranial EEG showed that spontaneous activations of the anterior insula precede phasic pupil dilations—a reliable index of transient increases of arousal[9]. While our study did not manipulate arousal or tonic alertness, in a previous study using the same object recognition task and concurrent MEG and eye-tracking[94], we showed that pupil-linked arousal fluctuations induce a pattern of perceptual behavior changes highly similar to the behavioral effects of CO network prestimulus activity observed here (including a more liberal criterion, higher hit rate, higher categorization accuracy, and no effect on FAR). Using whole-head MEG decoding, this earlier experiment also revealed an MEG correlate of pupil-linked arousal in the low-frequency range, termed the non-content-specific (NCS) activity[95]. Higher prestimulus NCS activity predicted a more liberal criterion on the recognition task with no influence on d′, as well as enhanced discrimination accuracy (see Fig. 4 in ref. 95)—a pattern of behavioral effects mirroring those of CO network prestimulus activity uncovered herein. Therefore, it is tempting to speculate that the CO network might be the underlying generator of the NCS spontaneous activity previously discovered—an intriguing possibility that awaits future investigation.

A limitation of our study is the lack of temporal resolution to investigate potential dynamic changes and inter-areal communication in the prestimulus period[96–98]. Future investigation using intracranial recordings can help fill this important gap. In addition, while our findings have illuminated key facets of how prestimulus activity within distributed brain networks impacts different aspects of perceptual behavior and their underlying mechanisms, future studies employing intervention-based techniques such as transcranial magnetic stimulation or intracranial stimulation will allow the direct testing of causal relationships between the identified prestimulus activity and perceptual behavior. In addition, we did not collect respiratory, visceral, or cardiac signals, which might influence perceptual behavior[99,100]. Future research could incorporate these physiological measures to investigate whether they have predictive influences on conscious perception in the prestimulus period and, if so, what are the involved brain mechanisms. Finally, in the present study, we have shown how spontaneous activity influences post-stimulus processing, which is likely to contain both the perceptual and decisional processes. Future research designed to minimize the need for active decision-making, such as using a no-report paradigm[101], can help further disentangle the mechanisms by which spontaneous brain activity influences perceptual and decisional processes, respectively.

In conclusion, our findings reveal the intricate mechanisms governing the impact of spontaneous brain activity on conscious object recognition and perceptual decision-making. We uncovered multiple cortical and subcortical regions whose prestimulus activity selectively influences different aspects of perceptual behavior, including criterion, sensitivity, and categorization. Our results further illuminate how prestimulus activity from these distributed brain regions shapes multiple aspects of stimulus-related processing, providing concrete mechanistic insights into these behavioral effects. Together, these findings contribute significantly to building a more comprehensive framework of the brain mechanisms underlying conscious perception that considers antecedent factors[35,38] and hold implications for understanding how the prevalent spontaneous activity contributes to a wide range of complex brain functions.

## Methods

### Participants
We recruited 38 volunteers (26 females, determined based on self-report; mean age 27 years, range 20–38 years) to participate in the fMRI experiment. The study protocol (#15-01323) was approved by the Institutional Board Review of New York University Grossman School of Medicine, and the experiment was conducted in accordance with the Declaration of Helsinki. Each participant provided written informed consent prior to the participation. All participants were right-handed, neurologically healthy, and had normal or corrected-to-normal vision. They received monetary compensation for their participation. Ten participants were excluded due to failing to complete the experiment and three were removed from data analysis due to poor behavioral performance, resulting in a final sample size of $n = 25$ for the analysis. The dataset analyzed here was previously published in a study that investigated the neural mechanisms underlying conscious object recognition in the post-stimulus time interval[40]. However, the data from the prestimulus interval analyzed in this study have not been previously investigated and the research questions addressed in this study are distinct from those in the earlier publication. Analysis based on sex or gender was not performed in this study because there were no sex or gender-specific hypotheses regarding the neural processing associated with the visual object recognition.

### Experimental procedure
The experiment involved two separate scanning sessions, which took place on two different days. During the first day, participants completed an image contrast staircase procedure during the anatomical MRI acquisition, as well as functional localizers. On the second day, participants performed an 8-min recognition task to confirm the threshold contrasts obtained on day one. If necessary, we conducted an additional 8-min staircase session to adjust for threshold changes across the two days. This was followed by the main object recognition task.

### Visual stimuli
The stimulus set used in this study included five distinct exemplars from four categories: face, house, man-made object, and animal. The images were obtained from public domain labeled photographs or from Psychological Image Collection at Stirling (PICS, http://pics.stir.ac.uk). The images were resized to 300 × 300 pixels, converted to grayscale, and their pixel intensities were normalized by subtracting the mean and dividing by the standard deviation. A two-dimensional Gaussian kernel with a standard deviation of 1.5 pixels and 7 × 7 pixels size was applied to each image. Additionally, scrambled images were generated by shuffling the phase (obtained via 2-D Fourier transform) of one randomly chosen exemplar from each category, resulting in a total of 20 real images and 4 scrambled images. The stimuli were presented using Psychophysics Toolbox (version 3) and MATLAB (R2017a) via an MRI-compatible screen (BOLDScreen, Cambridge Research System, 120 Hz frame rate) behind the MR scanner's bore opening. Participants viewed the stimuli via a mirror attached to the

MR head coil from a distance of ~210 cm, corresponding to a visual angle of ~8°.

Before the main task, participants were subjected to an adaptive staircase procedure "QUEST" to determine the image contrast that would result in a recognition rate of 50%. A more detailed description of the procedure is provided in ref. 40, 95.

## Main task

Participants completed an object recognition task consisting of 360 trials. Prior to each trial, there was a prestimulus period that lasted between 6 and 20 s, during which a gray background with a central fixation cross was presented. The trial started with the presentation of an image appearing at central fixation for 66.7 ms. The image intensity increased gradually from 0.01 to the threshold contrast intensity. After a delay ranging from 4 to 6 s, participants had to indicate the category of the presented image within two seconds by pressing a button. If they did not recognize the object, they were instructed to make a genuine guess. The stimulus-response mapping was indicated by the words face/house/object/animal presented in order randomized across trials. Subsequently, participants were asked to report their recognition (yes/no). They reported yes if they were able to see meaningful content, and no if they only see non-identifiable noise patterns. The main task was divided into 15 runs of ~7.5 min each. During each run, all 24 unique image exemplars (including 5 real and 1 scrambled exemplars for each object category) were presented once in a randomized order.

## Object category localizer

The details of the functional localizer scan have been described in previous publication[40]. Briefly, participants viewed the same set of real images in the main task, but presented at full contrast, and performed a one-back memory task. The localizer scan consisted of twenty blocks of object category presentation. Each block had a duration of 14 s and consisted of images from a single object category. Images were presented at a rate of 1 Hz, with a blank screen fixation of 500 ms in between. Participants were instructed to press a button whenever they noticed an image being presented twice consecutively. Each category block was repeated five times. The order of presentation was randomized, interspersed with 8 s blank fixation screens.

## Behavior analysis

Signal detection theory was applied to describe the recognition behavior. Hit rate (HR) was calculated as the proportion of real image trials wherein participants reported seeing an object. The false alarm rate (FAR) was calculated as the proportion of scrambled image trials wherein the participants reported seeing an object (even though the image did not contain an object). Following Macmillan and Kaplan correction[102], extreme values of HR and FAR (1 and 0) were substituted with $1 - \frac{1}{2N_{real}}$ and $\frac{1}{2N_{scr}}$, respectively. Here, $N_{real}$ denotes the number of trials featuring real images, while $N_{scr}$ represents the number of trials involving scrambled images. HR and FAR were then used to compute criterion (c) and sensitivity (d') as follows:

$$c = -\frac{1}{2}((Z(HR) + Z(FAR)) \tag{1}$$

$$d' = Z(HR) - Z(FAR) \tag{2}$$

where $Z$ is an inverse normal cumulative distribution function.

Furthermore, the categorization accuracy was determined as the proportion of trials in which participants reported the correct object category. For scrambled images, the categorization accuracy was determined based on the category of the original image.

## MRI scanning protocol

The MRI data were collected at the Center of Biomedical Imaging, NYU Langone, using a 7 T Siemens scanner (Erlangen, Germany) equipped with a NOVA 32-channel head coil. A T1-weighted structural volume was obtained for co-registration and spatial normalization purposes using an MPRAGE sequence with a voxel size of $1 \times 1 \times 1$ mm³, FOV of $256 \times 256$ mm², 176 slices, TR of 3000 ms, TE of 4.49 ms, and flip angle of 6°. Functional volumes were obtained using a T2*-weighted gradient echo-planar imaging (EPI) sequence that covered the whole brain ($2 \times 2 \times 2$ mm³, 0.2 mm gap, 54 slices, FOV = $192 \times 192$ mm², TR of 2000 ms, TE = 25 ms, flip angle = 50°, acceleration factor/GRAPPA = 2, multi-band factor 2). 227 volumes were obtained on each run of the main task and 221 volumes for the object localizer. Both the stimulus onsets in the main task as well as the presentation block onsets were temporally locked to the onset of volume acquisition.

## fMRI preprocessing

The fMRI data were preprocessed using the FSL software package (version 5.0.10, http://fsl.fmrib.ox.ac.uk/fsl/fslwiki/FSL). The EPIs were spatially realigned to the mean of the corresponding run using the six-parameter rigid body transformation to account for head motion. Time differences in slice acquisition were corrected, and the EPIs were coregistered to the participant's individual structural volume. Spatial smoothing was applied using a 3 mm and 4 mm full width at half maximum (FWHM) Gaussian kernel for the main task and functional localizer, respectively. Independent component analysis was employed to detect and eliminate artifacts caused by motion, arteries, or CSF pulsation. Finally, subject-level data were normalized to the Montreal Neurological Institute coordinate system (MNI) before being included in the analysis.

## Linear mixed-effects models (LMM)

In order to minimize the potential effects of sensory and motor responses on the prestimulus baseline data we first employed the general linear models (GLM) approach, implemented in FSL FEAT (fMRI expert analysis tool). A high pass filter with a cut-off frequency of 150 s was applied to remove slow drifts in the fMRI time series. In addition, prewhitening as implemented in the FSL FILM was performed to account for temporal autocorrelation in the data. For each run and participant, the preprocessed data were modeled with 17 predictors accounting for effects related to visual stimuli from 16 conditions (4 object categories ×2 image types ×2 recognition reports) and motor responses. These regressors were aligned to the onset of visual stimuli and onset of questions, respectively, and were each convolved with a gamma-shaped hemodynamic response function at the onsets of the respective events (half-width of 3 s and lag of 6 s). The resulting residuals corresponding to −2 to 0 s (1 TR) prior stimulus onsets, which were devoid of stimulus- and motor-related effects, were used as a proxy of spontaneous prestimulus activity. For each participant, residual maps corresponding to individual trials were spatially smoothed with a 5 mm FWHM Gaussian filter.

To examine whether recognition behavior changes with the amplitude of prestimulus spontaneous activity, we used LMMs as implemented in the stasmodels package (0.14.0)[103] in Python (version 3.8). For each voxel in each participant, trials were sorted into five groups based on the quintiles of their residual amplitude distribution. Four behavioral metrics (HR, FAR, d', and c) were computed for each trial group. The LMM included fixed effects for the trial group and intercept while treating participants as a random effect on the intercepts. The model was defined as follows:

$$Behavior_{i,G} \sim \chi * G + \chi_0 + \gamma_i * G + \gamma_{0i} + \varepsilon_{i,G} \tag{3}$$

In this model, $\chi$ denotes the fixed effect parameters that were shared by all participants, $\gamma_i$ represents the random effects parameter

for participant $i$, and $G$ represents the trial group (1–5) which the behavioral metrics were calculated for.

For each recognition behavior, the described model was fit to participants' data at each voxel, yielding a whole-brain coefficient map that indicated the strength of each voxel's prestimulus activity's influence on the specific variable of interest. A coefficient bigger than zero ($\chi > 0$) at a specific voxel indicates that the behavioral metric increases with prestimulus activity in that voxel. A coefficient smaller than zero ($\chi < 0$) suggests an inverse relation between prestimulus activity and the behavioral metric. The whole-brain map was subjected to a cluster-level inference using the Gaussian Random Field Theory, as implemented in FSL. We reported significant voxel clusters at $p < 0.05$, corrected for familywise error rate (FWE) at the cluster level with an initial cluster-defining voxel-wise threshold of $p < 0.01$.

For the analysis on categorization behavior, we used an ROI-based approach. For each participant, all trials were divided into five groups based on each voxel's prestimulus activity amplitude within a specific ROI, and categorization accuracy was calculated for each trial group. We then computed the mean categorization accuracy across voxels within each ROI, separately for each trial group. We then constructed an LMM model with the same parameters as described above to assess whether categorization accuracy was dependent on prestimulus activity in that ROI. The ROI-based analysis was separately conducted on four ROIs (see below). Statistical significance was reported at $p < 0.05$, FDR corrected across ROIs.

Additional LMMs were conducted to examine the potential effects of head motions during the prestimulus period on aspects of recognition behavior, including HR, FAR, d', and c. The relative framewise displacement between the two functional volumes preceding the stimulus onset, as output by FSL MCFLIRT, was utilized as an index to quantify the magnitude of head motion during the prestimulus period. This metric serves as an estimate of the extent to which the head position during the prestimulus time period of interest (−1 TR, −2 to 0 s prior to stimulus onset) deviated relative to the head position during the preceding TR (−4 to −2 s prior to stimulus onset). Following the LMM approach described earlier, trials were categorized into five groups based on the magnitude of the head motion during the prestimulus period, and behavioral metrics for each trial group were computed. Subsequently, LMMs were fit to examine whether the behavioral metrics changed as a function of prestimulus head motion magnitude.

### Definition of regions of interest

Four regions of interest (ROIs) were defined based on statistically significant criterion- and sensitivity-predictive clusters obtained from LMMs (Fig. 2, cluster-corrected $p < 0.05$). The vmPFC ROI was defined as the union of the criterion- and sensitivity-predictive clusters in the vmPFC and included a total of 1676 voxels. The CO network ROI encompassed voxel clusters that showed an inverse relation with the criterion and intersected with the salience network from the resting state network (RSN) parcellation[104] as well as the thalamus region from the 64-dimensional Dictionary of Functional Modes Atlas[105] (DiFuMo, region 47) and consisted of 762 voxels. The visual network ROI included criterion- and sensitivity-predictive clusters that intersect with the visual network from the RSN parcellation (1568 voxels). Lastly, the RSC was defined as the intersection of sensitivity-predictive clusters and RSC (region 111) from the 256-dimensional DiFuMo (146 voxels).

### Trial-to-trial variability analysis

For each subject, we extracted fMRI data recorded at the last TR before and the first two TRs after each stimulus onset. Since TRs were time-locked to the stimulus onsets, the extracted data corresponded to data recorded between −2 and 4 s relative to stimulus onsets. fMRI data were divided into two groups based on the prestimulus activity amplitude in a specific seed ROI, resulting in two trial groups

corresponding to high and low prestimulus activity trials. For each voxel at each TR in each trial group, we calculated the standard deviation across all trials and used it as a measure of trial-to-trial variability. This resulted in a whole-brain standard deviation map for each TR and trial group. To make group inference on the difference in trial-to-trial variability between high and low prestimulus activity trials at a given TR, subject-level whole-brain maps corresponding to the two trial groups were compared using a paired-sample t-contrast across the whole brain. Prior to the group inference, whole-brain maps were spatially smoothed with a 2 mm FWHM Gaussian kernel. This analysis procedure was repeated for each seed ROI. Statistical significance was reported at $p < 0.05$, corrected at cluster level for multiple comparisons with an initial cluster-defining voxel-wise threshold of $p < 0.01$.

### Stimulus-triggered responses conditioned by prestimulus activity

For each subject, fMRI data acquired from each run were divided into two trial groups based on the prestimulus activity amplitude in a specific ROI and fed into a GLM (150 s high-pass filter) using FSL Feat. We included two regressors to model brain responses to the visual stimuli from the two trial groups, each aligned to stimulus onsets. A third regressor was included in the GLM to account for motor responses and was aligned to the onset of each question in each trial. In addition, we incorporated nuisance regressors, comprising six movement parameters as well as the mean signals in the white matter and cerebrospinal fluid, as identified by FSL FAST with cut-off threshold of 0.9. All regressors were convolved with a gamma-shaped hemodynamic response function.

For each subject, we computed a contrast estimate between parameter estimates corresponding to high and low prestimulus activity trials in each run. The resulting contrast estimates were averaged across runs, yielding a mean contrast estimate per subject. To assess group-level effects, subject-level contrast estimates were tested against zero using a two-tailed t-contrast in FSL FLAME1. Cluster inference was performed using Gaussian Random Field Theory. We reported significant clusters at $p < 0.05$, with an initial cluster defining threshold of $p < 0.01$.

### Category decoding

Whole-brain searchlight decoding analysis was performed to assess whether the amount of category-related information in any brain region changes with the prestimulus activity level of each specific ROI.

The decoding models were trained using category localizer data, independently of the main task data which were used to test the models. The training data consisted of parameter estimates obtained from GLM applied to the object category localizer data. Each of the 20 object category presentation blocks was modeled separately with a predictor convolved with a gamma-shaped HRF. This resulted in five parameter estimate maps for each object category, serving as training samples in the decoding analysis.

The test set comprised main task data obtained from trials corresponding to a specific prestimulus activity level of a specific ROI. To this end, trials were median split into two groups according to the mean prestimulus activity amplitude across voxels within the ROI. Data obtained from each run were subjected to a GLM comprising predictors for all possible combinations between prestimulus activity level (high/low), recognition report (yes/no), and real image categories (face/house/animal/object). In addition, scrambled images and motor responses were modeled with separate predictors, yielding a total of 18 predictors. All predictors were convolved with gamma-shaped hemodynamic response functions at the onsets of the respective events. Parameter estimates for recognized real image categories from a specific prestimulus activity level were tested for accuracy of category prediction using the searchlight decoding models trained on the independent localizer data.

Searchlight decoding was performed on subject-level data using logistic regression models ($c = 1$) as implemented in the nilearn (0.9.2) scikit-learn (1.3.0)[106] package. We employed a 6-mm radius spherical searchlight and moved it voxel-by-voxel through the entire brain. At each voxel location, the decoder was trained to distinguish between object categories based on the patterns of BOLD responses within the searchlight during the functional localizer scan. The decoder was subsequently applied to response patterns corresponding to a specific prestimulus activity level in the main task. The decoders' prediction performance was evaluated using balanced decoding accuracy. This process was repeated for each voxel in each prestimulus activity level.

To perform group-level inference, we compared subject-level accuracy maps obtained from the two trial groups using a two-tailed paired $t$-test. The resulting statistical maps were subjected to a cluster-level inference using Gaussian Random Field Theory. The cluster significant threshold was set at $p < 0.05$ (two-tailed) with a cluster-defining threshold at $p < 0.01$. To avoid spurious findings, only voxels showing statistically significant above-chance decoding accuracy (chance level: 25%) at an uncorrected $p$-value of $< 0.05$ in at least one of the trial groups were incorporated into the final result.

### SDT simulation

To simulate how trial-to-trial variability influences the criterion and sensitivity, we conducted simulations using the SDT framework. To begin with, we generated a data set consisting of 10,000 observers, each completing 5000 trials of a recognition task with yes/no responses (2500 trials for each response outcome). To approximate the perceptual behavior observed in the empirical data, the internal response to target (representing real images) and nontarget (representing scrambled images) stimuli were drawn from two independent identical Gaussian distributions with a mean ($\mu$) of 2 and 0, respectively. The standard deviation of the Gaussian distribution ($\sigma$) was set to 3 for both target and nontarget stimuli. To align with the individually titrated HR of 0.5 that was observed throughout the study (see Results section and Fig. S1), the decision boundary that separated the recognition reports (yes vs no) was positioned at the mean value of the target distribution. The chosen simulation parameters produced highly similar behavior metrics (c: 0.33, d': 0.67) as in the empirical data (c: 0.35, d': 0.63).

We then repeated the simulations with decreasing standard deviations ($\sigma$). The values of $\sigma$ for both target and non-target distributions varied between 3 and 2 in steps of 0.2, while their means ($\mu$) remained unchanged. Since the HR was unaffected by the prestimulus vmPFC activity in the empirical data, the decision boundary was kept constant at the mean value of the target distribution across all $\sigma$ levels.

We assessed the effect of trial-to-trial variability on each of the behavioral metrics using linear regression. A positive regression coefficient ($\beta > 0$) indicates a given behavioral metric increases with increasing response variability, while a negative regression coefficient ($\beta < 0$) indicates an inverse relation. We reported statistical significance at $p < 0.05$ (FDR corrected for multiple comparisons).

### Reporting summary

Further information on research design is available in the Nature Portfolio Reporting Summary linked to this article.

## Data availability

This paper consists of analysis of previously published data. Unthresholded statistical maps generated in this study are available at https://neurovault.org/collections/17373/. Source data are provided with this paper.

## Code availability

We used publicly available open-source software toolboxes and custom scripts written in Python to analyze our data. Code supporting this study is available at a dedicated Github repository: https://github.com/BiyuHeLab/NatCommun_Wu2024

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

## Acknowledgements

This work was supported by U.S. National Institutes of Health/National Eye Institute (R01EY032085; PI: B.J.H.) and an Irma T. Hirschl Career Scientist Award (to B.J.H.).

## Author contributions

Y.W., E.P., and B.J.H. designed research; Y.W. and E.P. conducted analyses; M.L. provided analysis tools; Y.W. conducted visualization of results; Y.W. and B.J.H. wrote the paper.

## Competing interests

The authors declare no competing interests.
