## [Peer Review File · Nature Communications]

Network mechanisms of ongoing brain activity's influence on conscious visual perceptionREVIEWER COMMENTS

Reviewer #1 (Remarks to the Author):

Wu and colleagues studied the effects of prestimulus ongoing brain activity on perceptual decision-making and conscious recognition, using 7 Tesla fMRI and a visual perception task at threshold levels on 25 healthy participants. The authors found a diverse set of impacts on perceptual behavior by prestimulus ongoing activity originating from visual cortices and regions of the default-mode network (e.g., ventromedial prefrontal cortex and retrosplenial cortex) and cingulo-opercular network (e.g., the thalamus and anterior insula). The study further uncovered that the ongoing activity in each network distinctly influences various aspects of how stimuli are processed. This includes the magnitude of the evoked activity in both cortical and subcortical circuits, the variability of responses from trial to trial, and the encoding of stimulus content. For example, high prestimulus activity in the vmPFC was associated with higher sensitivity (d') and a more conservative criterion in the recognition task and reduced variability in distributed sensory-evoked activity. High prestimulus activity in the visual network was associated with a more conservative criterion in the recognition task and lower categorization accuracy, as well as higher neural variability and worse stimulus encoding. High prestimulus activity in the cingulo-opercular network was associated with a more liberal criterion in the recognition task and enhanced categorization accuracy. Simulation results provided further explanation on the ventromedial prefrontal cortex's role on recognition behavior including both sensitivity and criterion. For instance, a reduction of response variability can result in both enhanced sensitivity and a more conservative criterion.

I had the pleasure of reviewing this manuscript. This is a thorough work that provides an in-depth analysis of the interplay between prestimulus ongoing activity and perceptual processing. The manuscript is well-crafted, and the authors deserve commendation for their significant and enlightening findings. I have a few minor suggestions to offer, which I hope will further enhance the quality of the manuscript.

1. How much do the clusters that predict criterion and sensitivity overlap spatially? It would be interesting to see if these clusters can be represented on a separate map with different colors to highlight their shared and unique areas.
2. In Methods, independent component analysis was employed to detect and eliminate artifacts caused by motion, arteries, or CSF pulsation. However, the study does not clearly address whether factors like prestimulus head motion (e.g., measured by frame-wise displacement), cardiac, or respiratory signals might influence the perceptual outcomes. Is it possible to conduct control analyses to directly assess the impact of these factors?
3. For the linear mixed-effect models used in the study, the decision to categorize trials into five groups may raise some concern. It would be helpful if the authors could provide a rationale for this particular division. Additionally, it would be valuable to know if the results would remain consistent with a different number of trial groups, whether fewer or more.
4. In a previous study by van Vugt et al. (2018), it was found that prestimulus dlPFC activity is potentially related to perceptual criterion. The authors of this current work could beneficially discuss how their findings align or differ from van Vugt et al.'s, exploring both agreements and discrepancies in the context of prestimulus activity's influence on perception.

Zirui Huang

Reviewer #2 (Remarks to the Author):

Network mechanisms of ongoing brain activity's influence on conscious visual perception
Wu et al. – Nature Communications

This paper describes an extensive investigation into the correlation between pre-stimulus BOLD signal amplitude and both behavioral parameters and post-stimulus BOLD characteristics across the brain.

My main concern is related to the task that was used and, in particular, the analysis of behavior. The authors use a task in which observers view noisy non-scrambled images and scrambled images and subsequently categorize them into 4 categories, while the images are presented at a low contrast. The subjects also indicate whether they recognized the content, i.e. they made a subjective judgement about their own perceptual performance. It was unclear to me what exactly the two responses probe with regard to 'conscious visual perception'. What is 'recognized' in the recognition task?

This directly translates in confusion and concern about the classification of trials as hits, misses, false alarms and correct rejections. The recognition task appears to boil down to detecting something like 'objectness content'? Since 'yes' answers for scrambled objects are labeled as false alarms, this suggests that the scrambled images do not have whatever subjects are supposed to detect, whereas non-scrambled images do. In case of a trial with a non-scrambled picture, the subject thus has a hit if he/she reports 'recognition' and a miss if he/she reports 'no recognition'. Judging from this labeling the participants are thus recognizing whether an image is scrambled or not? This is consistent with the application of SDT, but confusing in how it relates to the research question.

There are at least three sources of confusion for me:

1. Some or most subjects were able to correctly categorize some of the scrambled images (as can be seen in Fig. 1E). Apparently, the scrambled pictures could be identified and thus also recognized. Given that the images were also shown at low contrast, it is unclear to me what the subjects were doing. Were they really trying to discriminate between scrambled and non-scrambled? What is the role of contrast here? Do low-contrast scrambled objects start to look like non-scrambled ones? Or is it possible that some of the subjects reported about their confidence of recognizing the category? Suppose that the subject recognizes that the shape of the scrambled object is a house, how discriminable is that from a non-scrambled low-contrast house? If the subject is confident that it is a house, he/she would presumably report that he/she recognized it, and not whether it was scrambled or not.

- What were the precise instructions to the subjects?

2. My impression from the paper is that the question asked to the subject is whether they recognized it, see Fig. 1C which states "Recognition: YES or NO". But the definition of a hit or miss in the analysis is not whether they recognized it but whether they could discriminate scrambled from non-scrambled images. Hence, if subject would report that they recognized a scrambled picture and say YES, this would count as a false alarm even though the subject was right.

3. What is the role of image categorization in the analysis? It remains unclear what happens to trials in which the image is put in the wrong category and the subject reports that he/she recognized it. Are these also classified as hits? Or misses? Or taken out of the analysis? My guess is that in classifying hits/misses/etc, the category question were completely ignored, but is this a valid approach?

In summary, I am concerned about the validity of this paradigm to probe conscious perception and I am also concerned about the definition of hits, misses, false alarms and correct rejections. I am unsure whether the subjects carried out a classification between scrambled/non-scrambled pictures (the basis of the authors definition of hits, misses, false alarms and correct rejections) or whether they were carrying out a recognition task.

To illustrate the impact on the measurement of d' : Imagine a very good observer who recognizes many of the scrambled images. According to the present analysis the d' of the observer should be very low, given the high "false alarm" rate, which could be close to the hit rate. This scenario is conceivable, given that some subjects are clearly above chance level for the scrambled images (Fig. 1E)

What could potentially be measured with this paradigm? There are 4 categories of trials for the normal images (1) subjectively recognized and correct (2) subjectively recognized and incorrect (3) subjectively not recognized and correct and (4) subjectively not recognized and incorrect. The same 4 categories are also there for the scrambled images (let us call them (5-8)). There may exist other ways to sort the trials and to thereby rescue this possibly valuable data set. E.g. one could compare recognized correct and incorrect trials, or one could compare correct trials that were subjectively recognized or not recognized.

Given that I have doubts about the validity of the behavioral classification of the trials and that these measures were subsequently used to sort the fMRI data, I don't think the present analysis of fMRI signals can support any conclusions about conscious visual perception.

We thank the reviewers for the careful review and thoughtful suggestions. We have performed additional analyses and thoroughly revised the manuscript to address these comments. Following the reviewers' suggestions, we have made the following changes to the figures:

- In response to Reviewer #1's point 1, an additional SI figure has been included (Fig S4), which shows the overlap between sensitivity- and criterion-predictive voxels.
- In response to Reviewer #1's point 3, a new SI figure has been included (Fig S5), which shows sensitivity- and criterion-predictive prestimulus activity resulting from linear mixed-effect models using 3 or 7 trial groups based on the prestimulus activity magnitude. These results are similar to our previous LMM results using 5 trial groups.
- Figure 1E has been revised to better distinguish the interpretation of categorization performance for real and scrambled images, and substantial changes to the relevant text have been made to clarify the points raised by Reviewer #2.

Below we respond to each of the reviewers' comments in detail. Original comments from the reviewers are in blue and our replies are in black. Black font with indentation denotes quoted text from the revised manuscript.

Reviewer #1 (Remarks to the Author):

Wu and colleagues studied the effects of prestimulus ongoing brain activity on perceptual decision-making and conscious recognition, using 7 Tesla fMRI and a visual perception task at threshold levels on 25 healthy participants. The authors found a diverse set of impacts on perceptual behavior by prestimulus ongoing activity originating from visual cortices and regions of the default-mode network (e.g., ventromedial prefrontal cortex and retrosplenial cortex) and cingulo-opercular network (e.g., the thalamus and anterior insula). The study further uncovered that the ongoing activity in each network distinctly influences various aspects of how stimuli are processed. This includes the magnitude of the evoked activity in both cortical and subcortical circuits, the variability of responses from trial to trial, and the encoding of stimulus content. For example, high prestimulus activity in the vmPFC was associated with higher sensitivity (d') and a more conservative criterion in the recognition task and reduced variability in distributed sensory-evoked activity. High prestimulus activity in the visual network was associated with a more conservative criterion in the recognition task and lower categorization accuracy, as well as higher neural variability and worse stimulus encoding. High prestimulus activity in the cingulo-opercular network was associated with a more liberal criterion in the recognition task and enhanced categorization accuracy. Simulation results provided further explanation on the ventromedial prefrontal cortex's role on recognition behavior including both sensitivity and criterion. For instance, a reduction of response variability can result in both enhanced sensitivity and a more conservative criterion.

I had the pleasure of reviewing this manuscript. This is a thorough work that provides an in-depth analysis of the interplay between prestimulus ongoing activity and perceptual processing. The manuscript is well-crafted, and the authors deserve commendation for their significant and enlightening findings. I have a few minor suggestions to offer, which I hope will further enhance the quality of the manuscript.

We thank the reviewer for the positive comments and helpful suggestions.

1. How much do the clusters that predict criterion and sensitivity overlap spatially? It would be interesting to see if these clusters can be represented on a separate map with different colors to highlight their shared and unique areas.

In response to the reviewer's suggestion, we have now included a new Supplementary figure (Fig S4) illustrating the spatial overlap between sensitivity- and criterion-predictive clusters. These results are now described on page 6:

“To quantitatively assess which brain regions carried prestimulus activity predictive to both sensitivity and criterion, we overlaid the maps containing significant sensitivity- and criterion-predictive clusters (**Fig S4**). Within the vmPFC, the sensitivity- and criterion-predictive clusters had substantial overlap (572 voxels), accounting for 55.7% of the sensitivity-predictive and 47.6% of the criterion-predictive voxels. [...] In comparison, we also observed a small overlapping cluster of 55 voxels in the left V3 (**Fig S4**). The relatively small size of the overlap was unsurprising given the finding of a spatially very confined sensitivity-predictive cluster in the visual network (**Fig 2C**).”

the last paragraph on page 10 (newly added text underlined):

“Intriguingly, despite the robust influence on trial-to-trial variability in visual regions, prestimulus activity in the visual network was predominantly associated with a criterion effect and had no significant impact on sensitivity (except for a small cluster in V3).”

and the first paragraph on page 11

“Our results reveal a central role of vmPFC: Not only did the vmPFC contain the largest clusters of perceptually relevant prestimulus activity, which significantly influenced both sensitivity and criterion of recognition (**Fig 2B-C**), but it also stood out as the area with the strongest impact on both of these behavioral metrics across the whole brain (**Tables S1-S2**).”

2. In Methods, independent component analysis was employed to detect and eliminate artifacts caused by motion, arteries, or CSF pulsation. However, the study does not clearly address whether factors like prestimulus head motion (e.g., measured by frame-wise displacement), cardiac, or respiratory signals might influence the perceptual outcomes. Is it possible to conduct control analyses to directly assess the impact of these factors?

We have conducted an additional analysis to evaluate the potential impact of prestimulus head motion on perceptual outcomes including hit rate, false alarm rate, sensitivity and criterion. To this end, we adopted an LMM approach, similar to the approach employed to evaluate prestimulus brain activity's influence on recognition behavior. This analysis did not reveal any significant effects of prestimulus head motion on any of the behavioral metrics, indicating that the changes in perceptual behavior were not driven by the prestimulus head motion. The details of this analysis are described in “Linear Mixed effect models (LMM)” Methods section on page 22:

“Additional LMMs were conducted to examine the potential effects of head motions during the prestimulus period on aspects of recognition behavior, including HR, FAR, d' , and c . The relative framewise displacement between the two functional volumes preceding the stimulus onset, as output by FSL MCFLIRT, was utilized as an index to quantify the magnitude of head motion during the prestimulus period. This metric serves as an estimate of the extent to which the head position during the prestimulus time period of interest (-1 TR, -2 to 0 second prior to stimulus onset) deviated relative to the head position during the preceding TR (-4 to -2 second prior to stimulus onset). Following the LMM approach described earlier, trials were categorized into five groups based on the magnitude of the head motion during the prestimulus period and behavioral metrics for each trial group were computed. Subsequently, LMMs were fit to examine whether the behavioral metrics changed as a function of prestimulus head motion magnitude.”

The results of this analysis are now described in the second to last paragraph on page 6:

“Lastly, to control for the potential effects of head motion, we employed additional LMMs to evaluate whether head motion during the prestimulus period had any predictive influences on conscious recognition (see Methods for details). This analysis did not reveal significant prestimulus head motion effects on any of the behavioral metrics (HR: $\chi = 0.003$, $p = 0.574$; FAR: $\chi = 0.013$, $p = 0.144$; d' : $\chi = -0.026$, $p = 0.431$; c: $\chi = -0.023$, $p = 0.149$), rendering head motion an unlikely confounding factor.”

We acknowledge the potential possibility that prestimulus cardiac or respiratory signals might have influences on perceptual behavior. Unfortunately, we did not collect cardiac or respiratory data during this experiment because our 7T scanner is not equipped with such physiological monitoring capabilities. We have now added new text to the second to last paragraph of the discussion section on page 19 to address this limitation:

“In addition, we did not collect respiratory, visceral or cardiac signals, which might influence perceptual behavior^{99, 100}. Future research could incorporate these physiological measures to investigate whether they have predictive influences on conscious perception in the prestimulus period and, if so, what are the involved brain mechanisms.”

99. Azzalini D, Rebollo I, Tallon-Baudry C. Visceral signals shape brain dynamics and cognition. *Trends in cognitive sciences* **23**, 488-509 (2019).
100. Park H-D, Correia S, Ducorps A, Tallon-Baudry C. Spontaneous fluctuations in neural responses to heartbeats predict visual detection. *Nature neuroscience* **17**, 612-618 (2014).

3. For the linear mixed-effect models used in the study, the decision to categorize trials into five groups may raise some concern. It would be helpful if the authors could provide a rationale for this particular division. Additionally, it would be valuable to know if the results would remain consistent with a different number of trial groups, whether fewer or more.

To address this concern, we have now re-conducted the LMM analysis with 3 and 7 trial groups, while keeping all other parameters identical to the original analysis. The results, shown in a new SI figure (Fig S5), are nearly identical to those from the original analysis using 5 trial groups (Fig 2). These results are now described in the second to last paragraph on page 6:

“To validate the observed prestimulus activity’s effects on behavioral outcomes, we conducted analogous whole-brain LMMs using a different number of trial groups (3 or 7) to identify brain regions where prestimulus activity predicted sensitivity or criterion. As shown in **Fig S5**, the results from these analyses are nearly identical to those from the original analysis using 5 trial groups (**Fig 2**), suggesting the robustness of our findings.”

4. In a previous study by van Vugt et al. (2018), it was found that prestimulus dIPFC activity is potentially related to perceptual criterion. The authors of this current work could beneficially discuss how their findings align or differ from van Vugt et al.’s, exploring both agreements and discrepancies in the context of prestimulus activity’s influence on perception.

Following the reviewer’s suggestion, we have expanded our discussion regarding the commonality and discrepancies between the dIPFC finding in Vugt et al. (2018) and our result on page 17 and 18 of the discussion section:

“Interestingly, a previous monkey neurophysiological study using a threshold-level detection task

concluded that prestimulus neuronal firing rates in the dIPFC predicts detection criterion³¹. By contrast, using whole-brain fMRI in humans, we did not find any sensitivity- or criterion-predictive prestimulus activity in the dIPFC (**Fig 2**). Other than differences in species and recording modalities, differences in the task paradigms likely contributed to this discrepancy. While the monkeys in that study detected a low-level visual target (circle) presented in the periphery, in our study, human subjects detected the presence of a meaningful visual content (e.g., a house). Therefore, these two studies targeted visual detection vs. object recognition, respectively. In addition, the authors of Ref³¹ suggested that the dIPFC neurons they recorded from were involved in eye movement planning, so they may be especially relevant for visual detection in the periphery. By contrast, our stimuli were presented at fovea.”

Reviewer #2 (Remarks to the Author):

Network mechanisms of ongoing brain activity’s influence on conscious visual perception
Wu et al. – Nature Communications

This paper describes an extensive investigation into the correlation between pre-stimulus BOLD signal amplitude and both behavioral parameters and post-stimulus BOLD characteristics across the brain.

My main concern is related the task that was used and, in particular, the analysis of behavior. The authors use a task in which observers view noisy non-scrambled images and scrambled images and subsequently categorize them into 4 categories, while the images are presented at a low contrast. The subjects also indicate whether they recognized the content, i.e. they made a subjective judgement about their own perceptual performance. It was unclear to me what exactly the two responses probe with regard to ‘conscious visual perception’. What is ‘recognized’ in the recognition task?

We appreciate the reviewer's comments and acknowledge that there was insufficient clarity in the description of the task utilized in our study, which led to various points of confusion. This was partly because the same data set was used in a previous publication focusing on stimulus-triggered brain responses (Levinson et al., *Nat. Commun.* 2021, “Cortical and subcortical signatures of conscious object recognition”), as mentioned in Methods, *Participants* (P. 19) and the 2nd paragraph in the Results section (P. 3). The same task paradigm with slightly different trial-level timing was also used in a previous MEG study from our lab (Podvalny et al., *Nat. Commun.* 2019, “A dual role of prestimulus spontaneous neural activity in visual object recognition”). Therefore, we were not as detailed and thorough as we should have been in describing the task paradigm in the Results section, and we sincerely apologize for that.

To answer the reviewer’s question, in our recognition task, participants were asked whether they saw a meaningful content in the visual stimulus, rather than making a subjective judgment about their own perceptual performance. That is, the recognition question was tailored to capture individuals' subjective experiences of recognizing coherent objects. This was previously described in the Results section (P. 3):

“For the recognition question, participants were instructed to respond “yes” whenever they saw a meaningful object in the image, even if the object appeared unclear and noisy, but to respond “no” if they saw nothing or only low-level features such as lines or cloud-like abstract patterns.”

In addition, we have expanded on the following sentence: (newly added text underlined)

“As such, this second question probed the success or failure of conscious object recognition rather than the conscious detection of low-level image features (for additional details, see ⁴⁰), aligning with the established definition in prior studies on object recognition^{41, 42, 43}.”

40. Levinson M, Podvalny E, Baete SH, He BJ. Cortical and subcortical signatures of conscious object recognition. *Nature communications* **12**, 2930 (2021).
41. Grill-Spector K, Kushnir T, Hendler T, Malach R. The dynamics of object-selective activation correlate with recognition performance in humans. *Nature neuroscience* **3**, 837-843 (2000).
42. Fisch L, *et al.* Neural “ignition”: enhanced activation linked to perceptual awareness in human ventral stream visual cortex. *Neuron* **64**, 562-574 (2009).
43. Bar M, *et al.* Cortical mechanisms specific to explicit visual object recognition. *Neuron* **29**, 529-535 (2001).

Therefore, our recognition question strictly probes subjective object recognition, instead of metacognition/confidence. Although these two concepts are related and often correlated, they are dissociable and distinct processes of consciousness (e.g., see Dehaene et al., Science 2017). Please also see our responses below for further clarifications.

This directly translates in confusion and concerned about the classification of trials as hits, misses, false alarms and correct rejections. The recognition task appears to boil down to detecting something like ‘objectness content’? Since ‘yes’ answers for scrambled objects are labeled as false alarms, this suggests that the scrambled images do not have whatever subjects are supposed to detect, whereas non-scrambled images do. In case of a trial with a non-scrambled picture, the subject thus has a hit if he/she reports ‘recognition’ and a miss if he/she reports ‘no recognition’. Judging from this labeling the participants are thus recognizing whether an image is scrambled or not? This is consistent with the application of SDT, but confusing in how it relates to the research question.

The reviewer is correct that only the real (non-scrambled) images contained meaningful content, while the scrambled ones did not. Each scrambled image was created by randomly shuffling the phase of a real, non-scrambled image from one of the four object categories. Despite retaining low-level features of the original images (which have different statistical properties between categories), the resulting scrambled images lacked any meaningful content. Therefore, a ‘yes’ response to a scrambled image in the recognition task indicated that the participant subjectively perceived a meaningful content in the image despite the lack of any such content, constituting a false alarm.

It is crucial to emphasize that the participants' goal in the recognition task did not involve discriminating between real and scrambled images (see our instruction to the subjects quoted in the response to the point above). In fact, participants were not informed about the presence of scrambled images.

As such, instead of evaluating the physical presence of a stimulus, our participants' goal in the recognition task was to report whether they subjectively recognized a meaningful content in the images. Consequently, both hits and false alarms in this recognition task signaled the subjective perception of an object, with hits reflecting recognizing an object in real images, and false alarms signaling the false perception of an object in scrambled images. With this operationalization, we assessed what signals surpassed the threshold for conscious recognition of coherent objects, rather than the detection of low-level features.

We have now incorporated new text to clarify these points on page 3: (newly added text underlined)

“Crucially, our stimuli set included both real and scrambled images (**Fig 1B**). Scrambled images were generated by phase-shuffling a randomly selected real image from each category to preserve low-level features that differ between categories while removing any meaningful content, and were presented at the same contrast as the corresponding original image. Participants were not informed about the inclusion of scrambled images. Scrambled image trials thus served as catch trials to evaluate the participant’s general tendency to report recognition of

a meaningful content.”

There are at least three sources of confusion for me:

1. Some or most subjects were able to correctly categorize some of the scrambled images (as can be seen in Fig. 1E). Apparently, the scrambled pictures could be identified and thus also recognized.

We apologize for the confusion here, which was partly due to insufficient clarity in our previous manuscript. The phase scrambling procedure destroys any meaningful content in the image, while preserving low-level features whose statistics differ between categories. Thus, objectively, these pictures could not possibly be identified or recognized. We believe that the false alarm responses (responding “Yes” to a scrambled image) reflect genuine false perceptions, when subjects perceive a meaningful content despite being presented with a meaningless noise input. Such false perceptions in healthy subjects have been reported by multiple previous papers [e.g., (Chalk et al., 2010; Haarsma et al., 2023)].

This interpretation of the false alarms responses, as false perceptions instead of mistaken button presses (errors in the motor output), was motivated by our analysis of the participants’ responses in the categorization task. Here, to shed light on what’s going on in the false alarm trials, we scored the categorization responses according to the category of the *original* image used to generate the scrambled image. E.g., if the participant answered ‘animal’ when the original image was a dog, it was scored as correct. Following this procedure, we found that the ‘categorization accuracy’ in false alarm trials (61.7%) was substantially higher than the chance level (25%), suggesting that the low-level features preserved in the phase-scrambling procedure biased subjects’ false perceptions to be in the same category. As a concrete example, the scrambled images created from animal and face pictures have more rounded features than those created from object or house pictures; faced with these scrambled pictures, participants likely experience false perception of animals/faces more often than objects/houses.

We have made multiple revisions to clarify these points:

1) Fig 1E has been revised: the y-axis label was changed from “%correct” to “categorization accuracy (%)”. The trial types (Hit, Miss, FA, CR) are now labeled along the x-axis. And the real and scrambled image results are now broken into two disconnected sub-panels.

2) Further clarifications about how the categorization accuracy was computed for scrambled images are now included in the figure legend to Fig 1E (P. 5, top).

3) We have expanded on the relevant Results section (PP. 3-4): (newly added text underlined)

“To understand the nature of FAR trials (constituting 28% of scrambled image trials), in which participants answered “Yes” to the recognition question despite the image input being devoid of any meaningful content, we analyzed the categorization response patterns in these trials. Because the phase scrambling procedure preserved the statistics of low-level features that differed between image categories⁴⁸, we scored the accuracy of the categorization responses according to the original images used to generate the scrambled images (e.g., if the participant answered ‘animal’ when the original image was a dog, it was scored as correct). Categorization accuracy for ‘recognized’ scrambled image (false alarm) trials was $61.7 \pm 2.8\%$. It was significantly higher than that in ‘unrecognized’ scrambled image (correct rejection) trials ($27.7 \pm 2.2\%$, $W = 231$, $p = 5.96 \times 10^{-5}$), and significantly above the chance level ($W = 253$, $p = 3.95 \times 10^{-5}$). Thus, low-level features that differed between categories contributed to participants’ categorization responses on the false-alarm trials, suggesting that the false alarm responses likely reflect genuine false perceptions of meaningful content rather than incorrect button presses⁴⁰.”

Given that the images were also shown at low contrast, it is unclear to me what the subjects were doing. Were they really trying to discriminate between scrambled and non-scrambled? What is the role of contrast here? Do low-contrast scrambled objects start to look like non-scrambled ones?

As mentioned earlier, participants were not instructed to discriminate between real and scrambled images; instead, their task was to report their subjective recognition experience of a meaningful content. The contrast of each image was titrated for the individual participant to ensure that real images were presented at the threshold level (i.e., ~50% recognition rate), while scrambled images were presented at the same contrasts as the corresponding real images. The threshold-level presentation allowed us to achieve significant behavioral divergence (conscious recognition vs. lack thereof) under the same physical input, providing sufficient statistical power to investigate the influences of pre-stimulus spontaneous brain activity on perceptual outcome. Please see our response below for additional examples of using the SDT approach to investigate conscious perception in similar paradigms.

As both real and scrambled images were presented at the threshold level with a brief duration of 66 ms and shared low-level visual properties, they might appear similar. But again, scrambled images did not contain any meaningful content. Consequently, reporting subjective recognition of a meaningful object in response to a scrambled image constituted a false alarm and likely false perception.

Or is it possible that some of the subjects reported about their confidence of recognizing the category? Suppose that the subject recognizes that the shape of the scrambled object is a house, how discriminable is that from a non-scrambled low-contrast house? If the subject is confident that it is a house, he/she would presumably report that he/she recognized it, and not whether it was scrambled or not.

- What were the precise instructions to the subjects?

We hope that the above responses have clarified these issues. Briefly, for the categorization task, the subjects were instructed to report the object category in the presented image, regardless of whether they had recognized an object in that image. In cases of unrecognized images, they were instructed to make a genuine guess. For the recognition task, the subjects were instructed to report whether they subjectively recognized an object in the image. Subjects were instructed to respond “yes” whenever they saw a meaningful object, even if the object appeared unclear, and respond “no” when they saw nothing or low-level features only, such as lines or cloud-like abstract patterns. (For further details, see pages 3 and 20 of the manuscript.)

To directly address the reviewer’s question, false perception of a scrambled house image might, subjectively, be quite similar to the perception of a real, threshold-level house image, at least in some cases. And again, the subject’s task was to report their subjective recognition status veridically, instead of discriminating between real and scrambled images.

As mentioned earlier, despite the close link between confidence and conscious perception, they are distinct and dissociable processes (Dehaene et al., 2017; Rosenthal, 2019; Morales and Lau, 2021). Since we only asked participants to report their recognition experience but not their judgement of their decision, we are not able to specifically investigate the role of confidence in this task.

2. My impression from the paper is that the question asked to the subject is whether they recognized it, see Fig. 1C which states “Recognition: YES or NO”. But the definition of a hit or miss in the analysis is not whether they recognized it but whether they could discriminate scrambled from non-scrambled images.

Hence, if subject would report that they recognized a scrambled picture and say YES, this would count as a false alarm even though the subject was right.

The reviewer is correct in that the question posed to the subjects in the recognition task was indeed about whether they recognized a meaningful content in the image. Contrary to the reviewer's impression, however, the definition of a hit or miss in our analysis did not hinge on the subjects' ability to discriminate scrambled from non-scrambled images (please also see our responses to previous points). As illustrated in **Fig 1C**, hits/misses referred to whether they reported recognizing meaningful content in real images, while false alarms/correct rejections were based on their responses to scrambled images. Please also see our previous publications (Podvalny et al., 2019; Podvalny et al., 2021) for similar SDT analyses applied to the same task paradigm in an MEG experiment.

The scrambled images, by design, did not contain an object stimulus or any meaningful content. Therefore, a “yes” response to a scrambled image would always be objectively *incorrect*, and therefore constitutes a “false alarm”. That is, the subject cannot be right when answering “yes” to a scrambled image, even if their categorization response is scored as ‘correct’ when compared to the category of the original image used to generate that scrambled image. It simply means that the low-level features in the scrambled image biased the subject’s false perception to be in the same category.

3. What is the role of image categorization in the analysis? It remains unclear what happens to trials in which the image is put in the wrong category and the subject reports that he/she recognized it. Are these also classified as hits? Or misses? Or taken out of the analysis? My guess is that in classifying hits/misses/etc, the category question were completely ignored, but is this a valid approach?

The reviewer is correct in that our assessment of recognition-related responses (e.g., hits and misses) did not take into account the categorization responses. This is well aligned with the SDT framework, in which detection and discrimination performance are typically separately analyzed (Green and Swets, 1966). In fact, many classic studies only collected detection or discrimination responses, where a joint analysis of them is not possible. Here, we collect both detection and discrimination responses on every trial, consistent with previous studies on perceptual awareness [e.g., (Lau and Passingham, 2006; Del Cul et al., 2007; Hesselmann et al., 2011; Li et al., 2014; Samaha et al., 2016; Benwell et al., 2017)]. In these studies, the predominant approach is to consider awareness (as measured by a detection-type task) and objective performance (as measured by a discrimination task) separately, and analyze the neural activity related to them separately. As such, the responses to the awareness task are sorted into hits, misses, FA, and CR without consideration of the discrimination task response. A minority of studies, including Hesselmann et al., 2011 and Li et al., 2014 (work from our lab), analyzed the trials using the conjunction of detection and discrimination responses; however, this approach has significant shortcomings that we will discuss later.

It is possible to use 2-dimensional SDT modeling to simultaneously analyze detection and discrimination responses (King and Dehaene, 2014). However, although this approach can be illuminating in the analysis of behavior and has been fruitfully applied in the metacognition field (e.g., (Peters and Lau, 2015)), we are not aware of studies that have incorporated 2D SDT modeling into the analysis of neural data. In addition, the extension of 2D SDT modeling to a 4-alternative choice question, as in our case, is far from straightforward (Churchland and Ditterich, 2012).

We would also like to highlight that our study also probed how prestimulus brain activity influences categorization behavior. We show that prestimulus activity in both the visual network and CO network influenced categorization behavior, but in opposite directions (**Fig 6**), and that this pattern of findings is consistent with visual network’s modulation of stimulus encoding (**Fig 5**) and CO network’s tight relation to the modulation of tonic alertness (see discussion in the last paragraph on page 18). Together, our

findings provide a coherent picture of how prestimulus brain activity influences both the detection of object information in perceptual awareness (i.e., subjective object recognition) and the discrimination of object category information (i.e., objective categorization performance).

In summary, I am concerned about the validity of this paradigm to probe conscious perception and I am also concerned about the definition of hits, misses, false alarms and correct rejections. I am unsure whether the subjects carried out a classification between scrambled/non-scrambled pictures (the basis of the authors definition of hits, misses, false alarms and correct rejections) or whether they were carrying out a recognition task.

As clarified earlier, subjects carried out an objective recognition task that directly probed their subjective perception of a meaningful object, and the classification of trials into hits, misses, FA and CR was based on the subjective recognition responses. This approach is very standard in the study of conscious perception/perceptual awareness, and many previous studies have used detection-based SDT approach to analyze responses related to conscious perception, whether it's perception of low-level images (e.g., Gabor or shape) [e.g., (Limbach and Corballis, 2016; lemi et al., 2017)] or high-level images (e.g., objects) [e.g., (Podvalny et al., 2019; Podvalny et al., 2021)]. An elegant review on this topic can be found in (Samaha et al., 2020).

To illustrate the impact on the measurement of d-prime: Imagine a very good observer who recognizes many of the scrambled images. According to the present analysis the d' of the observer should be very low, given the high "false alarm" rate, which could be close to the hit rate. This scenario is conceivable, given that some subjects are clearly above chance level for the scrambled images (Fig. 1E)

We hope that our responses above have clarified this point. Briefly, such an observer would indeed have a lower d' due to the high false alarm rate. The high false alarm rate is justified, because the scrambled images are devoid of any meaningful content; thus, responding "seeing a meaningful content" to them constitutes a false alarm, and likely indicates false perception. This observer would not have good performance at all, because of the high false alarm rate. The discrimination accuracy is (artificially) high because we scored the discrimination responses based on the category of the *original* image used to generate the scrambled image, and it simply means that low-level image features preserved during the phase-shuffling procedure biased the subject's false perception. Nevertheless, because the scrambled image does not contain any meaningful object, this artificial "discrimination accuracy" is only used to shed light on what the subject might be perceiving in these false perception cases, instead of providing a genuine measure of discrimination performance. We hope that our revision to the text on pages 3-4 mentioned above has now cleared these confusions.

What could potentially be measured with this paradigm? There are 4 categories of trials for the normal images (1) subjectively recognized and correct (2) subjectively recognized and incorrect (3) subjectively not recognized and correct and (4) subjectively not recognized and incorrect. The same 4 categories are also there for the scrambled images (let us call them (5-8)). There may exist other ways to sort the trials and to thereby rescue this possibly valuable data set. E.g. one could compare recognized correct and incorrect trials, or one could compare correct trials that were subjectively recognized or not recognized.

As mentioned earlier in our response to the reviewer's point 3, we have indeed previously adopted this trial-sorting approach to combine the analysis of subjective awareness and objective performance (Li et al., 2014). A few other studies have done so as well (e.g., Hesselmann et al., 2011). However, our

impression is that the field has largely shifted away from this approach, and a predominance of studies now analyze neural activity related to subjective awareness and objective performance separately, often adopting an SDT approach [e.g., (Limbach and Corballis, 2016; Iemi et al., 2017; Podvalny et al., 2019; Samaha et al., 2020)]. This shift is well motivated by theoretical and methodological considerations, as explained below.

First, and central to our question about neural activity predicting subjective perception, the comparison between correct (or incorrect) trials that are subjectively recognized vs. not recognized is highly problematic. This is because %correct is drastically different between recognized and unrecognized trials (Fig 1E, 79% for hit trials and 32% for miss trials). Therefore, in correct, recognized trials and correct, unrecognized trials, the strength of the signal related to objective discrimination is vastly different. Hence, this comparison, thought to isolate neural activity related to awareness while controlling for that related to objective performance, is highly confounded by neural signals related to objective performance. (As an extreme example, if miss trials had 25% correct rate, it could still be sorted into correct and incorrect trials, but the correct trials would have no signal related to discrimination, given the chance-level discrimination performance.)

Second, such a contrast between subjectively recognized vs. not recognized trials confounds two different aspects of perceptual decision-making identified by SDT: sensitivity change vs. criterion change. In other words, the subject could be recognizing a real image due to a higher sensitivity (d') or a lower criterion (c), which imply entirely different mechanisms. Indeed, being able to dissociate d' and c is the primary motivation for the inclusion of catch (scrambled image) trials in our study, as well as in many previous studies. By adopting an SDT framework, we are able to separately pinpoint prestimulus activity that shifts the criterion or influences the sensitivity, which has important theoretical and mechanistic implications. This general approach was also adopted in our previous MEG work (Podvalny et al., 2019; Podvalny et al., 2021) on this topic, using an identical paradigm except for different trial-level timing.

Given that I have doubts about the validity of the behavioral classification of the trials and that these measures were subsequently used to sort the fMRI data, I don't think the present analysis of fMRI signals can support any conclusions about conscious visual perception.

We hope that our responses above have clarified the misunderstandings about our paradigm and apologize again for the insufficient clarity in the earlier version of the manuscript. We have implemented multiple revisions in the text to clarify these points, which has significantly improved the quality of the manuscript.

We would like to thank both reviewers again for the thorough review and helpful comments.

References

- Benwell CSY, Tagliabue CF, Veniero D, Cecere R, Savazzi S, Thut G (2017) Prestimulus EEG Power Predicts Conscious Awareness But Not Objective Visual Performance. *eNeuro* 4.
- Chalk M, Seitz AR, Series P (2010) Rapidly learned stimulus expectations alter perception of motion. *J Vis* 10:2.
- Churchland AK, Ditterich J (2012) New advances in understanding decisions among multiple alternatives. *Curr Opin Neurobiol* 22:920-926.

- Dehaene S, Lau H, Kouider S (2017) What is consciousness, and could machines have it? *Science* 358:486-492.
- Del Cul A, Baillet S, Dehaene S (2007) Brain dynamics underlying the nonlinear threshold for access to consciousness. *PLoS Biol* 5:e260.
- Green DM, Swets JA (1966) *Signal Detection Theory and Psychophysics*. New York: John Wiley & Sons.
- Haarsma J, Deveci N, Corbin N, Callaghan MF, Kok P (2023) Expectation Cues and False Percepts Generate Stimulus-Specific Activity in Distinct Layers of the Early Visual Cortex. *J Neurosci* 43:7946-7957.
- Hesselmann G, Hebart M, Malach R (2011) Differential BOLD activity associated with subjective and objective reports during "blindsight" in normal observers. *J Neurosci* 31:12936-12944.
- Iemi L, Chaumon M, Crouzet SM, Busch NA (2017) Spontaneous Neural Oscillations Bias Perception by Modulating Baseline Excitability. *J Neurosci* 37:807-819.
- King J-R, Dehaene S (2014) A model of subjective report and objective discrimination as categorical decisions in a vast representational space. *Philosophical Transactions of the Royal Society B: Biological Sciences* 369:20130204.
- Lau HC, Passingham RE (2006) Relative blindsight in normal observers and the neural correlate of visual consciousness. *Proc Natl Acad Sci U S A* 103:18763-18768.
- Li Q, Hill Z, He BJ (2014) Spatiotemporal dissociation of brain activity underlying subjective awareness, objective performance and confidence. *J Neurosci* 34:4382-4395.
- Limbach K, Corballis PM (2016) Prestimulus alpha power influences response criterion in a detection task. *Psychophysiology* 53:1154-1164.
- Morales J, Lau H (2021) Confidence tracks consciousness. *Qualitative consciousness: themes from the philosophy of David Rosenthal*:1-21.
- Peters MA, Lau H (2015) Human observers have optimal introspective access to perceptual processes even for visually masked stimuli. *eLife* 4:e09651.
- Podvalny E, King LE, He BJ (2021) Spectral signature and behavioral consequence of spontaneous shifts of pupil-linked arousal in human. *eLife* 10.
- Podvalny E, Flounders MW, King LE, Holroyd T, He BJ (2019) A dual role of prestimulus spontaneous neural activity in visual object recognition. *Nature communications* 10:3910.
- Rosenthal D (2019) Consciousness and confidence. *Neuropsychologia* 128:255-265.
- Samaha J, Sprague TC, Postle BR (2016) Decoding and Reconstructing the Focus of Spatial Attention from the Topography of Alpha-band Oscillations. *J Cogn Neurosci* 28:1090-1097.
- Samaha J, Iemi L, Haegens S, Busch NA (2020) Spontaneous Brain Oscillations and Perceptual Decision-Making. *Trends Cogn Sci* 24:639-653.

REVIEWERS' COMMENTS

Reviewer #1 (Remarks to the Author):

The authors have made superb revisions. I'm satisfied that my concerns have been completely resolved.

Reviewer #3 (Remarks to the Author):

Wu et al. study how spontaneous brain activity influence the processing of visual sensory stimuli and perceptual decision making. This topic is significant because spontaneous activity may involve non-linear interactions with sensory/cognitive/perceptual events, thus predicting behavioral, cognitive, and perceptual outcomes. While this topic has been previously studied (the authors share some of this previous work in the introduction and discussion sections), spontaneous brain activity/pre-stimulus activity is far less examined in the field than evoked/post-stimulus activity.

The authors implement a previously published at-threshold, object categorization perception task. Several brain regions were identified as predictive of task criterion and sensitivity for determining the object categories, including insula, LOC, OFC, fusiform, and vmPFC. Pre-stimulus activity in these regions revealed unique interactions with behavioral metrics on the object perception task. The authors summarize these regions into ROIs/networks: vmPFC ROI, visual, CO, and RSC networks. A simulation was also implemented to test the authors' hypotheses.

Overall, the methods are sound, the results are intriguing, and the manuscript is written clearly and offers strong points of discussion and interpretation of their findings. I believe the manuscript meets the high standards of research in this field. In addition, I share similar thoughts/critiques as offered by the other reviewers. Wu et al. adequately responded to those points of feedback. I also have some additional points for the authors to consider:

(1) The task delay interval (4-6s; the authors reported up to 4s) is too short to fully resolve the temporal profile of evoked hemodynamics (the canonical HRF with a peak time of 5-6s). This short post-stimulus interval limits interpreting the post-stimulus BOLD dynamics. This might be an important limitation because there are differences at TR 1 versus TR 2 (Figure 3), including changes in the precuneus, V1, and posterior parietal cortex that are present at TR 2 but absent at TR 1. These differences could represent some underlying neural dynamic unfolding overtime. Alternatively, as BOLD signal magnitude increases towards a peak near 5-6s in many (but not all) brain regions, some areas may only appear in analyses at later time points. I would be interested to hear if the authors believe there could be brain areas statistically subthreshold at 2- or 4-seconds post-stimulus but may appear at >6s?

(2) The results might be influenced by the engagement in a behavioral task that involves introspecting on conscious perception. For example, a key area of emphasis in this study is the vmPFC. The authors interpret their results in vmPFC as indicative that the DMN might be involved in conscious object recognition (lines 217-221). However, the administered task requires behavior/introspection on conscious experience of the kind that may fit the common narrative of DMN as an internally oriented network. Accordingly, the vmPFC results might be present when participants are judging and reporting on their subjective experiences in a perceptual decision-making task, but absent or involved in a different way when different task demands are required, or no task at all.

(3) The spontaneous brain activity might be modulated by engaging with a task. For example, the pre-stimulus interval may not involve the same kind of spontaneous activity as recorded in a no task/resting state study because arousal/vigilance level may be modulated as participants are anticipating a stimulus presentation and engaging in the behavioral requirements. If true, the spontaneous brain activity at rest (no task) might be different than that of a pre-stimulus interval of this or any other perception task. Likewise, in the discussion section, the authors hint of

possible task-based demands influencing their findings when they compare their results to another study: "Other than differences in species and recording modalities, differences in the task paradigms likely contributed to this discrepancy." The possible influence of task on spontaneous activity and post-stimulus dynamics does not invalidate the current results, but the authors might consider clarifying the scope of the results within the context of a task or argue why their findings are likely to be present in a task-independent setting.

(4) The authors write, "The results of this analysis, plotted in Fig S6 (-2-0 s period), reveal a very different pattern from that in the post-stimulus period, suggesting that the results in Fig 3 are mainly driven by stimulus-related processing." However, there appears to be many similarities between -2-0s and 0-2s for the vmPFC and visual network ROI. If the authors agree there are similarities, then their interpretation that the results are predominantly driven by stimulus-related processing might be tempered, instead driven by a combination of pre and post-stimulus processing. If the authors believe it is important to highlight pre and post-stimulus influences, showing an overlay map (like Figure S4) between -2-0s and 0-2s may help highlight what is similar and different at these different time points. I will leave it to the authors' discretion on the usefulness of this visualization approach.

(5) I noticed in Figure 2B and Figures S2/S5 there are clusters in TPJ. I do not believe the authors mention this area in the main manuscript, nor is it included among the four ROIs/networks. The authors do highlight the supramarginal gyrus, but the SMG cluster pointed out in Figure S3 seems more posterior to the cluster that I am calling TPJ seen in Figure 2B (lateral view, left hemisphere). TPJ might be relevant to interpret among the other highlighted clusters because this area has been shown to be involved in object perception (e.g., Nestmann et al., Neuroimage, 2021).

(6) Minor point: typo on Line 468: "Ihe"

We thank the reviewers for the careful review and thoughtful suggestions.

Below we respond to each of the reviewers' comments in detail. Blue font denotes quoted text from the reviewers' original comments and black font denotes our replies. Black font with indentation denotes quoted text from the revised manuscript.

Reviewer #1 (Remarks to the Author):

The authors have made superb revisions. I'm satisfied that my concerns have been completely resolved.

We thank the reviewer for the positive feedback.

Reviewer #3 (Remarks to the Author):

Wu et al. study how spontaneous brain activity influence the processing of visual sensory stimuli and perceptual decision making. This topic is significant because spontaneous activity may involve non-linear interactions with sensory/cognitive/perceptual events, thus predicting behavioral, cognitive, and perceptual outcomes. While this topic has been previously studied (the authors share some of this previous work in the introduction and discussion sections), spontaneous brain activity/pre-stimulus activity is far less examined in the field than evoked/post-stimulus activity.

The authors implement a previously published at-threshold, object categorization perception task. Several brain regions were identified as predictive of task criterion and sensitivity for determining the object categories, including insula, LOC, OFC, fusiform, and vmPFC. Pre-stimulus activity in these regions revealed unique interactions with behavioral metrics on the object perception task. The authors summarize these regions into ROIs/networks: vmPFC ROI, visual, CO, and RSC networks. A simulation was also implemented to test the authors' hypotheses.

Overall, the methods are sound, the results are intriguing, and the manuscript is written clearly and offers strong points of discussion and interpretation of their findings. I believe the manuscript meets the high standards of research in this field. In addition, I share similar thoughts/critiques as offered by the other reviewers. Wu et al. adequately responded to those points of feedback. I also have some additional points for the authors to consider:

We thank the reviewer for the positive evaluation of our work.

(1) The task delay interval (4-6s; the authors reported up to 4s) is too short to fully resolve the temporal profile of evoked hemodynamics (the canonical HRF with a peak time of 5-6s). This short post-stimulus interval limits interpreting the post-stimulus BOLD dynamics. This might be an important limitation because there are differences at TR 1 versus TR 2 (Figure 3), including changes in the precuneus, V1, and posterior parietal cortex that are present at TR 2 but absent at TR 1. These differences could represent some underlying neural dynamic unfolding overtime. Alternatively, as BOLD signal magnitude increases towards a peak near 5-6s in many (but not all) brain regions, some areas may only appear in analyses at later time points. I would be interested to hear if the authors believe there could be brain areas statistically subthreshold at 2- or 4-seconds post-stimulus but may appear at >6s?

The canonical hemodynamic response models with a peak at around 4–6 sec following an event were developed using strong, temporally extended sensory/motor stimuli and a spatial resolution of 3-4mm isotropic voxel size plus extensive spatial smoothing (Buxton et al., 1998; Friston et al., 2000). However, more recent evidence indicates that hemodynamic responses may be faster for brief and weak stimuli

(Polimeni and Lewis, 2021). For instance, Yeşilyurt et al. (2010) showed that the peak latency of BOLD responses to very short visual stimuli was shorter than 4 sec (see Figure R1 below). Furthermore, one often overlooked aspect is that, while the peak BOLD response is delayed, the initial response to a stimulus starts almost immediately—within a few hundred milliseconds (Yu et al., 2014). These early stages of the BOLD response already contain significant information (Menon et al., 1998). Given that our stimuli were weak (threshold-level contrast) and brief (66 ms), we reasoned that a substantial amount of the prestimulus activity's effects on post-stimulus processing could be captured within the reported 4-second time window.

Nevertheless, we acknowledge that there may be additional patterns beyond the reported time window. Due to the limited number of trials with the post-stimulus delay interval lasting longer than 4 seconds (less than half of all trials), we could not properly investigate BOLD responses at time points >4 sec after the stimulus onset due to the contamination by response-triggered activity at the single-trial level. Future studies using longer delay intervals are necessary to fully address this question. That said, given the reasons above, we believe that our 4-sec post-stimulus window should capture major effects of interest and be largely predictive of any additional effects.

Fig R1 Time courses of BOLD responses elicited by 5 ms visual stimuli with varying intensities delivered by a white light-emitting diode goggle, averaged across subjects ($n = 11$). Each color represents a particular stimulus intensity. Figure adapted from Yesiyurt et al. (2010).

(2) The results might be influenced by the engagement in a behavioral task that involves introspecting on conscious perception. For example, a key area of emphasis in this study is the vmPFC. The authors interpret their results in vmPFC as indicative that the DMN might be involved in conscious object recognition (lines 217-221). However, the administered task requires behavior/introspection on conscious experience of the kind that may fit the common narrative of DMN as an internally oriented network. Accordingly, the vmPFC results might be present when participants are judging and reporting on their subjective experiences in a perceptual decision-making task, but absent or involved in a different way when different task demands are required, or no task at all.

For several reasons, we believe that our vmPFC results are unlikely to be explained by a potential involvement in introspection. First, classic work ascribing an internally oriented function to the DMN typically employed internally oriented tasks, such as self-judgments (Gusnard et al., 2001), mind wandering (Mason et al., 2007), and semantic/episodic memory tasks (Shapira-Lichter et al., 2013). By contrast, our task is a classic “externally oriented” visual perception task, where participants only needed to report on what they see (or think they might have seen). The “recognized vs. unrecognized” question requires minimal introspection, but simply a veridical report of the visual recognition experience. Note that some in the field believe that there are strong relations between subjective experience (seen or recognized) and metacognition (i.e., confidence), where it could be said that confidence requires introspection. However, the fact that subjective perception and confidence are different and dissociable is well established (Dehaene et al., 2017; Rosenthal, 2019); moreover, metacognition has been linked to frontopolar and dorsal parietal regions that are distinct from the DMN (Kiani and Shadlen, 2009; Rahnev

et al., 2016).

Second, for any effects due to task-induced introspection, those are expected to occur after the primary perceptual processing—that is, in the post-stimulus period. However, a previous study from our lab using the same data set rigorously investigated post-stimulus activity related to subjective recognition and did not find significant effects centered on vmPFC (Levinson et al., 2021). Our present finding in the vmPFC concerns its prestimulus activity, which is difficult to account for by introspection-related processes.

Instead, we believe that our vmPFC finding is most easily explained by a potential role in top-down predictive processing. This interpretation is compatible with previous findings suggesting the vmPFC's role in generating predictive signals (Summerfield et al., 2006) as well as maintaining schematic information or cognitive maps in support of upcoming flexible behavior (Behrens et al., 2018). These considerations and related papers are discussed in detail in the Discussion section (P. 17).

(3) The spontaneous brain activity might be modulated by engaging with a task. For example, the pre-stimulus interval may not involve the same kind of spontaneous activity as recorded in a no task/resting state study because arousal/vigilance level may be modulated as participants are anticipating a stimulus presentation and engaging in the behavioral requirements. If true, the spontaneous brain activity at rest (no task) might be different than that of a pre-stimulus interval of this or any other perception task. Likewise, in the discussion section, the authors hint of possible task-based demands influencing their findings when they compare their results to another study: “Other than differences in species and recording modalities, differences in the task paradigms likely contributed to this discrepancy.” The possible influence of task on spontaneous activity and post-stimulus dynamics does not invalidate the current results, but the authors might consider clarifying the scope of the results within the context of a task or argue why their findings are likely to be present in a task-independent setting.

We agree that, in principle, task performance might modulate spontaneous brain activity. Yet, in order to probe spontaneous activity's influence on task functions, a task must be imposed, as otherwise there would be no function or behavior to measure. To mitigate the potential task-related influence on the spontaneous activity in our study design, we incorporated long and jittered inter-trial intervals (up to 20 seconds) with a roughly flat hazard rate (created by an exponential distribution of the ITIs). This approach aimed to reduce anticipatory effects. Furthermore, previous research has demonstrated that the spatiotemporal organization of spontaneous activity remains relatively stable across a wide variety of tasks (Cole et al., 2014; Gratton et al., 2018). This suggests that while task demands may influence specific aspects of brain activity, the underlying fluctuations in spontaneous activity and its overall organization are relatively robust. Given these considerations, we believe that the prestimulus activity measured in our study is likely to be representative of spontaneous brain activity more generally.

Nevertheless, this does not rule out the possibility that the influence of prestimulus activity on task-related neural responses and behavior might vary with the task demands. For example, compared to the task-related responses in our study, those in a no-report paradigm might lack a decision-related component. Therefore, we have included a discussion on this point: (P. 20)

“Finally, in the present study, we have shown how spontaneous activity influences post-stimulus processing, which is likely to contain both the perceptual and decisional processes. Future research designed to minimize the need for active decision-making, such as using a no-report paradigm¹⁰¹, can help further disentangle the mechanisms by which spontaneous brain activity influences perceptual and decisional processes, respectively.”

(4) The authors write, “The results of this analysis, plotted in Fig S6 (-2–0 s period), reveal a very different pattern from that in the post-stimulus period, suggesting that the results in Fig 3 are mainly driven by stimulus-related processing.” However, there appears to be many similarities between -2-0s and 0-2s for the vmPFC and visual network ROI. If the authors agree there are similarities, then their interpretation that the results are predominantly driven by stimulus-related processing might be tempered, instead driven by a combination of pre and post-stimulus processing. If the authors believe it is important to highlight pre and post-stimulus influences, showing an overlay map (like Figure S4) between -2-0s and 0-2s may help highlight what is similar and different at these different time points. I will leave it to the authors’ discretion on the usefulness of this visualization approach.

We agree with the reviewer’s observation that there are some similarities between -2-0s and 0-2s for the vmPFC and visual network ROI. In response to the reviewer’s comment, we have modified our interpretation of this result in the first paragraph on p11:

“The results of this analysis, plotted in Fig S6 (-2–0 s period), show that while there are some similarities between the prestimulus (-2–0 s) and poststimulus (0–2 and 2–4 s) periods, there are also substantial differences. Therefore, although some of the prestimulus activity’s effects on post-stimulus trial-to-trial variability might be inherited from the prestimulus period, the majority of the uncovered effects are likely driven by stimulus-triggered processing.”

(5) I noticed in Figure 2B and Figures S2/S5 there are clusters in TPJ. I do not believe the authors mention this area in the main manuscript, nor is it included among the four ROIs/networks. The authors do highlight the supramarginal gyrus, but the SMG cluster pointed out in Figure S3 seems more posterior to the cluster that I am calling TPJ seen in Figure 2B (lateral view, left hemisphere). TPJ might be relevant to interpret among the other highlighted clusters because this area has been shown to be involved in object perception (e.g., Nestmann et al., Neuroimage, 2021).

We thank the reviewer for highlighting the potential involvement of TPJ in our task and directing us to the relevant literature. The TPJ cluster seen in Figure 2B corresponds to the “L Angular Gyrus and Supramarginal Gyrus” in Table S1. This cluster was not included in further analysis due to its relatively small cluster size (it was the smallest cluster identified in this analysis). Furthermore, the effect in this brain area was less robust compared to those in other brain areas. For example, the criterion-related effect associated with the TPJ cluster, as observed in the main analysis (Fig 2B), could not be consistently reproduced when analysis parameter was altered (Fig S5). Nevertheless, all pertinent information regarding this cluster, including its cluster size, z statistic, as well as the peak coordinates, are documented in Table S1 for reference in future research. Because the paper cited by the reviewer investigated only post-stimulus activity, we did not see a strong connection with the present findings to warrant adding a discussion point.

(6) Minor point: typo on Line 468: “lhe”

Thank you. Corrected.

References

- Behrens TE, Muller TH, Whittington JC, Mark S, Baram AB, Stachenfeld KL, Kurth-Nelson Z (2018) What is a cognitive map? Organizing knowledge for flexible behavior. *Neuron* 100:490-509.
- Buxton RB, Wong EC, Frank LR (1998) Dynamics of blood flow and oxygenation changes during brain activation: the balloon model. *Magnetic resonance in medicine* 39:855-864.
- Cole MW, Bassett DS, Power JD, Braver TS, Petersen SE (2014) Intrinsic and task-evoked network architectures of the human brain. *Neuron* 83:238-251.
- Dehaene S, Lau H, Kouider S (2017) What is consciousness, and could machines have it? *Science* 358:486-492.
- Friston KJ, Mechelli A, Turner R, Price CJ (2000) Nonlinear responses in fMRI: the Balloon model, Volterra kernels, and other hemodynamics. *NeuroImage* 12:466-477.
- Gratton C, Laumann TO, Nielsen AN, Greene DJ, Gordon EM, Gilmore AW, Nelson SM, Coalson RS, Snyder AZ, Schlaggar BL (2018) Functional brain networks are dominated by stable group and individual factors, not cognitive or daily variation. *Neuron* 98:439-452. e435.
- Gusnard DA, Akbudak E, Shulman GL, Raichle ME (2001) Medial prefrontal cortex and self-referential mental activity: relation to a default mode of brain function. *Proc Natl Acad Sci U S A* 98:4259-4264.
- Kiani R, Shadlen MN (2009) Representation of confidence associated with a decision by neurons in the parietal cortex. *Science* 324:759-764.
- Levinson M, Podvalny E, Baete SH, He BJ (2021) Cortical and subcortical signatures of conscious object recognition. *Nature Communications* 12:2930.
- Mason MF, Norton MI, Van Horn JD, Wegner DM, Grafton ST, Macrae CN (2007) Wandering minds: the default network and stimulus-independent thought. *Science* 315:393-395.
- Menon RS, Luknowsky DC, Gati JS (1998) Mental chronometry using latency-resolved functional MRI. *Proc Natl Acad Sci U S A* 95:10902-10907.
- Polimeni JR, Lewis LD (2021) Imaging faster neural dynamics with fast fMRI: A need for updated models of the hemodynamic response. *Progress in Neurobiology* 207:102174.
- Rahnev D, Nee DE, Riddle J, Larson AS, D'Esposito M (2016) Causal evidence for frontal cortex organization for perceptual decision making. *Proc Natl Acad Sci U S A* 113:6059-6064.
- Rosenthal D (2019) Consciousness and confidence. *Neuropsychologia* 128:255-265.
- Shapira-Lichter I, Oren N, Jacob Y, Gruberger M, Hendler T (2013) Portraying the unique contribution of the default mode network to internally driven mnemonic processes. *Proc Natl Acad Sci U S A* 110:4950-4955.
- Summerfield C, Egnér T, Greene M, Koechlin E, Mangels J, Hirsch J (2006) Predictive codes for forthcoming perception in the frontal cortex. *Science* 314:1311-1314.
- Yeşilyurt B, Whittingstall K, Uğurbil K, Logothetis NK, Uludağ K (2010) Relationship of the BOLD Signal with VEP for Ultrashort Duration Visual Stimuli (0.1 to 5 ms) in Humans. *Journal of Cerebral Blood Flow & Metabolism* 30:449-458.
- Yu X, Qian C, Chen DY, Dodd SJ, Koretsky AP (2014) Deciphering laminar-specific neural inputs with line-scanning fMRI. *Nat Methods* 11:55-58.